# Stress Changes the Resting-State Cortical Flow of Information from Distributed to Frontally Directed Patterns

**DOI:** 10.3390/biology9080236

**Published:** 2020-08-18

**Authors:** Soheil Keshmiri

**Affiliations:** The Thomas N. Sato BioMEC-X Laboratories, Advanced Telecommunications Research Institute International (ATR), Kyoto 619-0237, Japan; soheil@atr.jp

**Keywords:** cortical flow of information, directed functional connectivity, stress, anxiety, transfer entropy

## Abstract

Despite converging evidence on the involvement of large-scale distributed brain networks in response to stress, the effect of stress on the components of these networks is less clear. Although some studies identify higher regional activities in response to stress, others observe an opposite effect in the similar regions. Studies based on synchronized activities and coactivation of these components also yield similar differing results. However, these differences are not necessarily contradictory once we observe the effect of stress on these functional networks in terms of the change in information processing capacity of their components. In the present study, we investigate the utility of such a shift in the analysis of the effect of stress on distributed cortical regions through quantification of the flow of information among them. For this purpose, we use the self-assessed responses of 216 individuals to stress-related questionnaires and systematically select 20 of them whose responses showed significantly higher and lower susceptibility to stress. We then use these 20 individuals’ resting-state multi-channel electroencephalography (EEG) recordings (both Eyes-Closed (EC) and Eyes-Open (EO) settings) and compute the distributed flow of information among their cortical regions using transfer entropy (TE). The contribution of the present study is three-fold. First, it identifies that the stress-susceptibility is characterized by the change in flow of information in fronto-parietal brain network. Second, it shows that these regions are distributed bi-hemispherically and are sufficient to significantly differentiate between the individuals with high versus low stress-susceptibility. Third, it verifies that the high stress-susceptibility is markedly associated with a higher parietal-to-frontal flow of information. These results provide further evidence for the viewpoint in which the brain’s modulation of information is not necessarily accompanied by the change in its regional activity. They further construe the effect of stress in terms of a disturbance that disrupts the flow of information among the brain’s distributed cortical regions. These observations, in turn, suggest that some of the differences in the previous findings perhaps reflect different aspects of impaired distributed brain information processing in response to stress. From a broader perspective, these results posit the use of TE as a potential diagnostic/prognostic tool in identification of the effect of stress on distributed brain networks that are involved in stress-response.

## 1. Introduction

Stress is a catalyst for many emotional disorders [1] that affect over 400 million individuals worldwide [2]. Its effect forces the brain into a state of fearful arousal that urges the need for rapid defense mechanisms [3,4,5]. It alters the brain functions by strengthening memories of stressful experiences [6,7,8]. It debilitates the brain capacity for reasoning and deliberation [9,10,11].

The brain response to stress involves large-scale dynamically interacting brain networks [12,13]. Among such networks, three appear to play a central role: the salience network (SN), the default mode network (DMN), and the fronto-parietal network (FPN) [14,15,16]. Research suggests that the irregularities in these three networks’ functions can lead to a wide range of (stress-related) psychiatric disorders [16].

Despite converging evidence on the involvement of these distributed networks in brain responses to stress [14,16,17], the effect of stress on their components is less clear [13,18]. For instance, considering the change in the brain activation, whereas Pruessner et al. [19] and Koric et al. [20] reported an elevated frontal activity in response to stress, Albert et al. [21] and Qin et al. [9] observed an opposite effect in this region. In the same vein, the use of such measures of coactivation as independent component analysis (ICA) [18,22,23] also yielded differing results. For instance, van Marle et al. [24], Veer et al. [25], and Vaisvaser et al. [26] found increased connectivity among components of these distributed brain networks in response to stress. On the other hand, Viard et al. [27], Zhang et al. [28], and van Oort et al. [29] reported their reduced connectivity.

However, these findings on the change in brain (de)activation in response to stress are not necessarily contradictory once one interprets the effect of stress on the brain in terms of its information processing capacity. Specifically, Wutz et al. [30] found that the modulation of information does not necessarily involve a change in local power. In other words, change in the brain’s information processing is not necessarily accompanied with a change in brain activity. In the same vein, Reid et al. [31] noted that associations among the brain regions in terms of correlation [27] or other measures of coactivation such as ICA [15,27,28,29] can arise in a variety of ways that may not relate to the extent of the influence among these cooccurring processes. As a result, they fall short of capturing a more comprehensive mapping between the observed associations and their underlying neural substrates [31,32,33]. Study of the effect of stress on these distributed networks in terms of information processing of their components can shed further light on the impact of stress on their inter-regional dynamics and their capacity for information-sharing and processing.

In this respect, the dynamical system analysis [34,35] frames the study of the brain function in terms of the interaction between its regions. Specifically, it treats the brain as a complex system [36,37] whose dynamics and ongoing activity [38] orchestrates its cognitive functions [39,40,41,42]. Among such approaches, Granger causality (GA) [43,44,45] and transfer entropy (TE) [46,47,48] are two effective measures of directed flow of information among distributed brain regions. Precisely, their strictly non-symmetric measure of the information exchange between different components of such brain networks helps verify whether the observed associations are indeed stemmed from causal (i.e., in its purely statistical term [49,50]) relations among these regions. An advantage of TE in comparison to GA is that whereas GA is based on linear vector autoregressive (VAR) [51,52] and hence linear in nature, TE is a nonlinear directional measure of flow of information [49,50].

In the present study, we investigate the utility of TE for quantification of the effect of stress on resting-state cortical information processing. For this purpose, we use the Max Planck Institute Leipzig Mind-Brain-Body Dataset [53]. This dataset pertains to 227 adults who completed a comprehensive set of neurophysiological (fMRI, EEG, cardiovascular measures, blood samples, and urine drug tests, etc.) and psychological tests (comprising 6 cognitive tests along with 21 questionnaires pertinent to personality traits and tendencies, eating, addictive, and emotional behaviors, etc.).

In our study, we consider the 216 subset of the total individuals who had their EEG data recorded [53]. To determine the participants’ stress-susceptibility, we use these 216 individuals’ responses to three psychological assessment questionnaires: the big-five of personality (a.k.a five-factor-model (FFM) [54]), the perceived stress questionnaire (PSQ) [55], and the state-trait anxiety inventory (STAI-G-X2) [56]. We then systematically identify 20 out of 216 participants who showed a significantly higher (hereafter, HIGH stress-susceptible) and lower (hereafter, LOW stress-susceptible) stress-susceptibility. Each of these HIGH and LOW stress-susceptible groups comprise 10 participants. Next, we use these 20 individuals’ resting-state multi-channel EEG recordings (both Eyes-Closed (EC) and Eyes-Open (EO) settings) and compute their distributed flow of information among different cortical regions using TE.

The contribution of the present study is three-fold. First, it identifies that the stress-susceptibility is characterized by the change in flow of information in fronto-parietal brain network. Second, it verifies that these distributed fronto-parietal regions that expand bi-hemispherically are sufficient to significantly differentiate between the HIGH and LOW stress-susceptible groups. In this regard, it also indicates that although these regions contribute differently to such distinctions, their differential contributions are rather non-significant. Third, it shows that in the case of HIGH stress-susceptible group, the flow of information between these distributed fronto-parietal brain regions is associated with a higher parietal-to-frontal flow of information.

The present results provide further evidence for the viewpoint in which the brain’s modulation of information is not necessarily accompanied by the change in its regional activity [30]. They further construe the effect of stress in terms of a disturbance that disrupts the flow of information among the brain’s distributed cortical regions. These observations, in turn, complement the previous research that primarily focused on the effect of stress on the brain networks in terms of synchronized activity among their components, thereby providing further insight towards understanding the causal relation (i.e., directional flow of information) among these brain regions [31,32,33]. From a broader perspective, these results posit the use of TE as a potential diagnostic/prognostic tool in identification of the effect of stress on distributed brain networks that are involved in the brain responses to stress. This observation becomes more intriguing considering the recent surge in application of machine learning and statistical frameworks to decoding of the brain activity [57,58,59,60].

## 2. Methods

Mind–body–brain dataset [53] provides a detailed description of the experimental settings, types of data collected, cognitive, personality traits, psychological questionnaires that the participants responded and more. In what follows, we summarize the information that pertain to the resting-state EEG recordings and the three psychological assessments that we used in this study.

### 2.1. Participants

Mind–body–brain dataset [53] included 227 participants in two age groups: the younger group (153 participants, 45 females, age: mean (M) = 25.1, median (Mdn) = 24.0, standard deviation (SD) = 3.1), and the older group (74 participants, 37 females, age: Mdn = 67.0, M = 67.6, SD = 4.7).

The resting-state EEG recordings which we used in the present study corresponded to 216 of these participants. Starting from 54th participant, the exact location of each 62 EEG electrodes were digitized using a Polhemus PATRIOT Motion Tracking System (Polhemus, Colchester, VT, USA) localizer and the Brainstorm toolbox [61]. This was done based on each participant’s head position relative to three fiducial points and the referenced electrode FCz. In the present study, we used this subset of 162 participants (i.e., 216 − 54 = 162 participants) (age: M = 38.61, Mdn = 30.0, SD = 20.14).

### 2.2. Resting-State EEG

#### 2.2.1. Acquisition

Mind–body–brain dataset [53] includes sixty-two-channel resting-state EEG recordings (61 scalp and a VEOG electrode below the right eye) from 216 human subjects. These channels were arranged according to 10–20 extended localization system, also known as 10-10 system [62]. They were referenced to FCz. During the recordings, EEG signals, per channel, per participant, were bandpass-filtered between 0.015 Hz and 1 KHz. They were further digitized at 2500 Hz sampling rate. Each EEG session, per participant, comprised 16 blocks. These blocks included two types of resting-state recordings: Eyes-Closed (EC) and Eyes-Open (EO). EC and EO each consisted of 8 blocks of length 60-s, per block. Every participants completed these two resting-state EEG recordings. For every participant, the EEG recording session started with EC (Figure 1A). During the experiment, the participants were seated in front of a computer screen and asked to stay awake. In the case of EO, they were asked to fixate their eyes on a black cross that was presented on the computer screen.

#### 2.2.2. Preprocessing

We used the preprocessed EEG recordings that were available through mind–body–brain dataset [53]. In this preprocessing pipeline, the raw EEG was first downsampled from 2500 Hz to 250 Hz. It was then, per channel, per participant, bandpass-filtered within 1–45 Hz using an eight-order Butterworth filter (i.e., four-order in both directions to minimize zero-crossing distortions by low-frequency drifts [63]). The data was then split into EC and EO, each comprising eight 60-s blocks. For each of these blocks, the EEG channels were visually inspected and the channels that were affected by such issues as frequent jumps/shifts in voltage and/or poor signal quality were rejected. During this step, data intervals that contained extreme peak-to-peak deflections or large bursts of high frequency activity were also removed (identified through visual inspection). Next, the dimensionality of the data (i.e., EEGs’ channel-dimension) was reduced by performing principal component analysis (PCA) and keeping PCs (≥ 30) that explained 95.0% of the variance. This step was then followed by independent component analysis (ICA) on temporal (i.e., EEG channels’ data points) dimension of data using the Infomax (runica) algorithm (step size: $\frac{0.00065}{log(number\;of\;channels)}$, annealing policy: when weight change > 0.000001, learning rate was multiplied by 0.98, stopping criterion maximum number of iterations 512 or weight change < 0.000001). Subsequently, components that reflected eye movement, eye blink, or heartbeat related artifacts were removed. Retained independent components for EO (M = 19.70, range = 9.0–30.0) and EC (M = 21.40, range = 14.0–28.0) conditions were then back-projected to the sensor space for further analysis. These analyses were performed using EEGLAB [64] (version 14.1.1b) for MATLAB (Delorme and Makeig, 2004).

In addition to these steps, we also detrended (using Matlab 2016a inbuilt detrend function) these EEG signals, per participant, per channel, per EC/EO, prior to any further computation and analysis.

### 2.3. EEG Channels Inclusion

The preprocessing steps that were applied on EEG recordings’ channel-dimension (i.e., PCA preprocessing step, Section 2.2.2) resulted in missing EEG channels in case of some of the participants. To balance the EEG channels for all participants, we therefore checked for the EEG channels that were common among all 20 HIGH and LOW stress-susceptible participants that were included in the present study. We found (Figure 1B) that 53 EEG channels were commonly available in all participants’ preprocessed EEG recordings which we used for our analyses. These channels were FP2, AF7, AF3, AFZ, AF4, AF8, C5, C3, C1, Cz, C2, C4, C6, CP5, CP3, CP1, CPZ, CP2, CP4, F5, F3, FZ, F1, F2, F4, F6, F8, FT7, FC5, FC3, FC1, FC2, FC4, FC6, FT8, P7, P5, P3, P1, PZ, P2, P4, P6, P8, PO9, PO7, POZ, PO3, PO4, PO8, and O1, OZ, O2.

### 2.4. An Overview of the Participants’ Selection and EEG Inclusion Process

Figure 2 illustrates the overall procedure through which participants and their respective EEG recordings for LOW and HIGH stress-susceptible groups were selected (see Appendix A for detailed information and results). This procedure comprised five following steps.

As a first step (Figure 2(1)), the 95.0% confidence intervals of the participants’ responses to each of NEO-FFI, PSQ, and STAI-G-X2 were calculated separately. Subsequently, those individuals whose responses to all of these three questionnaires fell in lower/upper boundary of respective confidence interval of its related questionnaire (indicated by green/red lines, respectively) were used to form the LOW/HIGH stress-susceptible groups.In the second step (Figure 2(2)), pair-wise TE values for all available 53-EEG-channels (Figure 1B), per LOW and HIGH stress-susceptible participants, per EC and EO settings’ matrices (i.e., 8 matrices, per setting) were calculated. This resulted in 8 TE matrices of size 53 × 53 (i.e., paired EEG channels’ TE values), per EC and EO settings. These 8 TE matrices were then averaged, per EC and EO settings, per participant, thereby yielding one averaged 53 × 53 TE matrix, per EC and EO settings, per participant.Third step (Figure 2(3)) included computing the 95.0% confidence interval for TE values, per EC and EO settings. For this purpose, TE values from the averaged EC and EO TE matrices of all participants in each LOW and HIGH stress-susceptible groups were separately combined (i.e., 10 × 53 × 53 = 28090 TE values, per EC and EO settings). Next, the TE values, per participant, per setting, that were below the upper boundary of their respective group’s confidence interval (i.e., per EC and EO settings) were discarded (i.e., set to zero).In step four (Figure 2(4)), we first counted the number of non-zero entries in each row of the averaged TE matrices, per individual, per EC and EO settings. Next, we combined the individuals’ counts for LOW and HIGH stress-susceptible groups separately and computed the 95.0% confidence intervals for these counts (i.e., per EC and EO settings, per stress-susceptible groups). We then discarded those EEG channels whose number of non-zero TE entries were below the upper boundary of their related confidence interval, per individual, per stress-susceptible groups, and per EC and EO settings.In step five (Figure 2(5)) we separately found the union of EEG channels among individuals in each of LOW and HIGH stress-susceptible groups, per EC and EO settings (Figure A2). In the case of EC, this step resulted in 18 EEG channels that were common between all participants in HIGH stress-susceptible group. These channels were AFZ, AF4, F1, F6, FT7, CZ, C2, C6, CP5, CP3, CP1, CP4, P7, P5, P4, P6, PO3, POZ. Similarly, there were 18 EEG channels that were common among all participants in LOW stress-susceptible group. They were FP2, AFZ, F3, F5, FZ, CP1, CP3, CP5, CPZ, P7, P3, P4, P2, P6, PO3, POZ, PO4, O2. We used the union of these EEG channels (without repetition) for comparative analyses between HIGH and LOW stress-susceptible groups in EC setting (Figure A2A). On the other hand, we found 21 EEG channels in EO setting that survived these thresholding steps and that were common between all participants in HIGH stress-susceptible group. Those channels were AF3, AF4, C1, C2, C6, CP3, CP4, CP5, F1, F2, F6, FC5, FC6, FT7, FT8, Fp2, Fz, O1, P1, P6, PO4. The number of such EEG channels in LOW stress-susceptible group was 17 (AF3, AF8, AFZ, C2, CP3, CPZ, F2, F4, FC5, FC6, FT7, Fp2, Fz, P2, PO7, PO8, PZ). Similar to the case of EC, we used the union of these EEG channels for comparative analyses between HIGH and LOW stress-susceptible groups for EO setting (Figure A2B).

### 2.5. Analysis

We performed three series of analyses: (1) analysis of the participants’ responses to neuroticism (NEO-FFI), worries (PSQ), tension (PSQ), and STAI trait anxiety (STAI-G-X2). (2) analysis of the participants’ total TEs (i.e., cumulative sum of TEs from one channel (e.g., AF4) to all the other 52 EEG channels). (3) analysis of the participants’ distributed TEs (e.g., AF4’s transferred information to each of the 52 channels, separately). Below, we elaborate on these analyses.

#### 2.5.1. Responses to Neuroticism (NEO-FFI), Worries (PSQ), Tension (PSQ), and STAI Trait Anxiety (STAI-G-X2)

To ensure the adequacy of neuroticism (NEO-FFI), worries (PSQ), tension (PSQ), and STAI trait anxiety (STAI-G-X2) for capturing the participants’ susceptibility to stress, we performed partial Spearman correlation on their responses to these questionnaires. We chose partial correlation to regress out any potential confounding effect that might have been present in their responses to each of these questionnaires (particularly between the two questions included from PSQ). We reported these results in three steps. First, we computed the partial Spearman correlations of all the 122 participants whose responses to neuroticism (NEO-FFI), worries (PSQ), tension (PSQ), and STAI trait anxiety (STAI-G-X2) were available. This ensured that these responses were indeed related with all the participants who participated in this study and whose EEG recordings were also available (Section 2.1). We reported these correlations in Section C.1. Second, we applied this partial Spearman correlation on the first subsets of 26 HIGH and 14 LOW stress-susceptible groups. These subsets included both female and male genders as well as younger and older participants. We reported these correlations in Section C.2. Last, we carried out these correlations on the final selection of HIGH (10 participants) and LOW (10 participants) male younger individuals. We reported these correlations in Section C.3.

To further validate the use of neuroticism (NEO-FFI), worries (PSQ), tension (PSQ), and STAI trait anxiety (STAI-G-X2) for choosing the participants in HIGH and LOW stress-susceptible groups, we also performed Wilcoxon rank sum test between every pairs of responses in these two groups (e.g., HIGH versus LOW responses to neuroticism). In addition, we verified these results using paired two-sample bootstrap test of significance (10,000 simulation runs) at 99.0% (i.e., p< 0.01) confidence interval. For the bootstrap test, we considered the null hypothesis “The difference between responses of HIGH and LOW stress-susceptible groups to a given questionnaire was non-significant” and tested it against the alternative hypothesis “The responses to a given questionnaire differed significantly between HIGH and LOW stress-susceptible groups.” We reported these results in Section C.4.

#### 2.5.2. Total TEs

For selected EEG channels of the participants in HIGH and LOW stress-susceptible groups, per EC and EO settings (Figure A2), we first computed their total TE as
(1)$$total\left(ch_{j}\right)=\sum_{ch_{i}}TE_{chi_{j}\to ch_{i}},\;\forall j\neq i$$
where *total* is a function that computes the sum of $j^{th}$ channel’s TEs to each of the other EEG channels, $ch_{i},\;i\neq j$ (Figure 1B). This computation is analogous to computing the sum of TEs for each row of 53 × 53 averaged TE matrices (i.e., one matrix per participant, per EC and EO settings). For example, total TE for channel AF4 was computed as sum of all TE values in its corresponding row entry in a given 53 × 53 averaged TE matrix. In essence, the total TE estimated the functional strength of each EEG channel. It is also worthy of note that since total TEs were computed by summing the row entries of the participants’ TE matrices, they quantified each EEG channels’ functional strength in terms of their total information transferred to every other EEG channel.

We then performed Wilcoxon rank sum test between each pair of selected channels for HIGH and LOW stress-susceptible groups (e.g., AF4 in HIGH and LOW). We carried out this analysis on EEG channels of both EC (Figure A2A) and EO (Figure A2B) settings. These tests identified several EEG channels that showed significantly different total TEs between HIGH and LOW stress-susceptible participants. In the case of EC, these channels (9 EEG channels in total) were in frontal (FP2, AF4, F1, F6), centroparietal (CP3 and CP4), parietal (P2 and P3), and occipital (O2) regions. For EO (14 EEG channels in total), they were in frontal (AF4, F1, F4), frontotemporal (FT7), central (C1 and C6), centroparietal (CP3, CPZ, and CP4), parietal (P1 and PZ), parieto-occipital (PO7 and PO8), and occipital (O1) regions.

To realize the importance of each of these EEG channels in distinguishing between HIGH and LOW stress-susceptible groups, we performed General Linear Model (GLM) [65,66] analysis on them. We opted for GLM analysis using logistic regression with sigmoid function. The inputs to this model were N × M matrices where N refers to the number of participants (i.e., 20 in our case) and M is the number of channels whose total TEs significantly differed between HIGH and LOW groups (i.e., 9 and 14 in the case of EC and EO, respectively). We used 1 and 0 as class labels for HIGH and LOW stress-susceptible groups, per EC and EO settings. We then carried out ANOVA analyses on this model’s coefficients (i.e., weights), per EC and EO settings. We used Matlab 2016a (“fitlm” and its corresponding “anova” functions) for these analyses. We reported the results of EC analyses in the main manuscript. We provided the results pertinent to EO setting in Section B.1 and Section B.2.

Last, we computed the Spearman correlations between the channels with significantly different total TEs between HIGH and LOW stress-susceptible participants and their responses to neuroticism (NEO-FFI), worries (PSQ), tension (PSQ), and STAI trait anxiety (STAI-G-X2) questionnaires. We reported these results in Section D.1 (in the case of EC) and Section D.2 (in the case of EO).

#### 2.5.3. Distributed TEs

We used the EEG channels with significantly different total TEs, per EC (9 channels: FP2, AF4, F1, F6, CP3, CP4, P2, P3, O2) and EO (14 channels: AF4, F1, F4, FT7, C1, C6, CP3, CPZ, CP4, P1, PZ, PO7, PO8, O1) and examined whether their distributed transfer of information also differed between HIGH and LOW stress-susceptible groups. For each of these channels (e.g., AF4), we combined its separate TEs to the remaining 52 EEG channels (i.e., the directed transfer of information to the other channels) for all of the participants, per HIGH and LOW group. It is worthy of note that the total TE of each of these channels (Section 2.5.2) was the cumulative sum of these separate TEs to each of the other 52 EEG channels. We then performed Wilcoxon rank sum test between each pair of these channels’ distributed TEs for HIGH and LOW stress-susceptible groups (e.g., AF4 in HIGH and LOW).

Last, we counted the number of regions that each of these channels transferred information to in HIGH and LOW stress-susceptible groups, per EC and EO settings, and applied Wilcoxon rank sum test on them to determine whether there was any significant difference between them. We reported the results pertinent to EC in the main manuscript. We provided the EO setting’s results in Section B.3.

#### 2.5.4. TE Computation, Effect Sizes and Bonferroni Correction

We used the TE implementation in [67]. For Kruskal–Wallis, we reported the effect size $r=\sqrt{\frac{\chi^{2}}{N}}$, as suggested by Rosenthal and DiMatteo [68], where $\chi^{2}$ and N are the test-statistics and the sample size, respectively. In the case of Wilcoxon test, we used $r=\frac{W}{\sqrt{N}}$ [69] as effect size with *W* denoting the Wilcoxon statistics and *N* is the sample size. The effect sizes of these tests are considered [70] small when *r* ≤ 0.3, medium when 0.3 < *r* < 0.5, and large when *r* ≥ 0.5. For ANOVA analysis of GLM’s coefficients (i.e., model’s weights), we reported $\eta^{2}$ effect size. In this case, the effect is considered [71] small when $\eta^{2}\leq$ 0.01, medium when 0.06 $\geq \eta^{2}$, and large when $\eta^{2}\geq$ 0.14. All the results reported are Bonferroni corrected ($\frac{0.05}{2}$ = 0.025) where 2 refers to HIGH and LOW stress-susceptible groups.

## 3. Results

### 3.1. Total TEs

Channel-wise paired Wilcoxon rank sum test identified nine EEG channels (Figure 3) whose total TEs were significantly different between HIGH and LOW stress-susceptible groups. These channels were FP2, AF4, F1, F6, CP3, CP4, P2, P3, and O2.

Among these channels, four channels (AF4, F1, F6, and CP4) showed significantly higher total TEs in HIGH stress-susceptible group (Table 1). On the other hand, the remaining five channels (FP2, CP3, P2, P3, and O2) were associated with significantly higher total TEs in LOW stress-susceptible group (Table 2). All these significant differences were associated with strong effect sizes.

### 3.2. GLM Analysis of the Channels with Significantly Different Total TEs

ANOVA analysis of the GLM coefficients (i.e., model’s weights) identified (Figure 4) non-significant difference (Table 3) of the contribution of each channel individually to predict the participants’ group membership (coefficients’ statistics: M = −0.0058, SD = 0.032, CI_95.0%_ = [−0.0731 0.05269]).

### 3.3. Distribution of the Information Transferred by the Channels with Significantly Different Total TEs

Similar to the case of total TEs, distributed TEs from AF4, F1, F6, and CP4 to other EEG channels were significantly higher among HIGH stress-susceptible group (Table 4).

We also observed that the distributed TEs of the same remaining five channels in the case of total TE values (FP2, CP3, P2, P3, and O2) were significantly higher among LOW stress-susceptible groups (Table 5).

On the other hand, contribution of these channels’ distributed TEs to other channels differed substantially between HIGH and LOW stress-susceptible groups. Figure 5 illustrates the distributed contribution of TEs from AF4, FP2, F1, F6, CP3, CP4, P2, P3, and O2 to the other channels whose significant differences were presented in Table 4 and Table 5. This figure indicates that whereas transfer of information from these channels primarily contributed to the frontal regions in the case of HIGH stress-susceptible group (Figure 5A), their contribution was more globally distributed among fronto-parietal in the case of LOW stress-susceptible group (Figure 5B).

Figure 6 and Figure 7 visualize the cortical regions which each of these channels with significantly different distributed TEs transferred information to. More specifically, these figures show the directed transfer of information (i.e., TE) from (i.e., out-degree) AF4, FP2, F1, F6, CP3, CP4, P2, P3, and O2 to the other cortical regions in the case of HIGH and LOW stress-susceptible groups. A comparison between these figures clarifies the substantially frontal-oriented distributed TE among HIGH stress-susceptible group (Figure 6) and its more distributed nature among fronto-parietal regions in the case of LOW stress-susceptible individuals (Figure 7). The more dense networks of TEs in the case of LOW versus HIGH stress-susceptible groups is apparent in these two figures. Wilcoxon rank sum test identified that the number of regions that each of these channels transferred information to was significantly higher in the case of LOW versus HIGH stress-susceptible participants (*p* = 0.000, *W*(16) = −3.59, *r* = 0.85, M${}_{HIGH}$ = 13.11, SD${}_{HIGH}$ = 1.36, M${}_{LOW}$ = 33.78, SD${}_{LOW}$ = 1.09). This difference was associated with a strong effect size.

## 4. Discussion

In the present study, we investigated the potential effect of stress on cortical flow of information using transfer entropy (TE) [46,47,48]. Our results identified a distributed cortical network that expanded bilaterally. This was in line with the previous findings that indicated the involvement of large-scale dynamically interacting brain regions in response to stress [12,14]. The use of TE also extended the previous research that primarily focused on the synchronized activity among components of such networks [15,17,27,28,29]. Precisely, TE’s strictly non-symmetric measure of information exchange facilitated the quantification of the dynamics of the information-sharing among different brain regions.

Van Oort et al. [13] observed that the use of different types of experimentally induced psychological and physical stressors by most of previous studies could potentially yield differential impacts on the brain response to stress [15,20,72]. To realize the common stress-induced effects on the brain distributed networks [14], they further asserted that such variations must be dissociated from more (potentially) general patterns. Some investigators indeed addressed this issue by using the resting-state brain activity that was collected before and after administration of such psychological and physical stressors [24,25,26,28]. Our study extended these previous attempts for attenuation of the effect of stressors on the observed changes in the brain networks by considering the resting-state that was not associated with any physical or psychological stressors. Specifically, the participants in the present study only completed the multidimensional mood state (MDBF) questionnaire [73] prior to their resting-state EEG recordings in which they ranked their moods (5-point Likert scale, from 1 (not at all) to 5 (very much) [53]). In this regard, it was interesting to note that the subtle difference in participants’ resting with their eyes closed or open (Section B.3) resulted in an apparent change in the pattern of cortical flow of information. This observation further highlighted the potential (confounding) impact that the experimental setting (i.e., in addition to the effect of different stressors [13]) can exert on the induced pattern of brain activity.

We observed that the regions with significantly higher total TEs were associated with the brain’s fronto-parietal network [29] bilaterally. Furthermore, the distribution of these regions appeared to highlight the cortical components of default mode network (DMN) [74,75,76]. This observation was in line with the previous findings that indicated the involvement of this network in brain’s stress-response [13,27,28,29]. These results extended the previous analyses that identified that the brain’s distributed emotion-specific activity may provide maps of internal states that correspond to specific subjectively experienced emotions [77,78,79]. Interestingly, although these regions contributed differently to such distinctions, their differential contributions were rather non-significant. This was in accord with Watson and Tellegen [80] that showed the joint activity from multiple regions best discriminated between different internal states. In this regard, Zhang et al. [81] also presented a high classification accuracy of pre- versus post-stress using the resting-state functional connectivity of healthy individuals. Our findings extended their results in two ways. First, our results provided evidence for the significant effect of stress on brain functional connectivity in the absence of any explicit stressor. Second, our results indicated that the effect of stress resulted in a change in brain’s distributed network whose flow of information substantially differed between individuals with low versus high stress-susceptibility.

In the case of HIGH stress-susceptible group, we observed that the information from these fronto-parietal regions flew to a substantially smaller number of other cortical regions. This observation was in accord with van Oort et al. [29] report on a reduced connectivity in fronto-parietal network in response to stress. It was also in line with Sheline et al. [82] and Kaiser et al. [83] who considered such a reduced connectivity to reflect an impaired ability to suppress attention to internal emotional states. our results further complemented these previous findings by extending them from synchrony between these regions to the case in which the change in transfer of information between different regions could be more evidently apprehended. It also (at least partially) addressed the issues concerning the insufficiency of the regional synchronization for a more comprehensive understanding of their level of associations [31,32,33].

The brain fronto-parietal network appears to act as a domain-general network [84,85]. It plays a pivotal role in a variety of brain functions [86,87] that range from self-referential processing [88], autobiographical memory [89], and emotion [90] to social cognition and theory of mind [91], decision making [92], and working memory [93]. In view of these observations, it is plausible to construe the observed reduction of the flow of information among the components of this network among HIGH stress-susceptible individuals to highlight the potential impact of stress on their overall brain cognitive ability for handling various personal and social life events [1,94]. This interpretation finds further evidence in the stress-induced dampening of higher-level cognitive functions [14,16].

We also observed that the fronto-parietal flow of information was predominantly projected onto the frontal cortex in the case of HIGH stress-susceptible individuals. This observation resonated with Cohen and D’Esposito [95] that reported an increase connectivity between frontal and parietal regions as a function of task difficulty. In this respect, our result might suggest the taxing effect of stress on the brain frontal regions of these HIGH stress-susceptible individuals to maintain its overall cognitive and behavioral control. This interpretation becomes more plausible considering the interfering effect of stress on prefrontal cortex that is necessary for the flexible control of behavior [11,96]. It may also reflect the self-criticism by these individuals whose impact is most pronounced in the frontal regions [97]. In this respect, the long-range flow of information from parietal-to-frontal regions could be explained in terms of the selective role of parietal cortex in cognitive processes that support individuals’ emotional distancing [98].

Our results suggest that some of the differences in the previous findings perhaps reflect different aspects of impaired distributed brain information processing in response to stress. For instance, in the case of increased [19,20] versus decreased [9,21] brain activity in response to stress, our results identified that the higher/lower flow of information within the same cortical lobe (e.g., frontal cortex) was not a distinct characteristic of HIGH or LOW stress-susceptible groups. Although a higher flow of information in one subset of the same cortical lobe’s regions was associated with HIGH stress-susceptible group, its other subset exhibited such a higher flow of information in the case of LOW stress-susceptible group. In the same vein, our results indicated that whereas LOW stress-susceptible group was characterized with a flow of information that was substantially more distributed, the HIGH stress-susceptible group exhibited the flow of information that was evidently focused on the frontal cortical regions. Such a variation in cortical information integration may lead to differing findings on increased [25,26] or decreased [27,28,29] brain regional synchronization if the direction of such influences were not accounted for [31,32,33]. This observation is evident in desynchronization effect of stress on various frontocortical regions that results in impaired ability to shift attention [99,100].

From a broader perspective, our results posit the potential use of TE as a diagnostic marker of the stress. For instance, TE can help identify the individuals that are at higher risk of stress-related neuropsychological disorders. Subsequently, it may also prove useful for tracking the effect of stress-related treatments on these patients through quantification of the changes in the pattern of information-sharing among their distributed cortical regions in comparison with healthy individuals with lower stress susceptibility.

## 5. Limitations and Future Direction

Recent studies [13] underlined three essential networks that are particularly involved in stress-response: the default mode network (DMN), the salience network (SN), and the central executive network (CEN) [16]. Although our results hinted at the fronto-parietal cortical components of such networks, the use of EEG in the present study did not allow for identification of the subcortical structures that are involved in these brain networks [14,15,101]. In this respect, the 162 subset of participants from Babayan et al. [53] that were included in our study had localized EEG channels. In addition, Babayan et al. [53] also provided the fMRI recordings of the individuals who participated in their study. This provides the future research with the opportunity to further our results by extending the use of TE to the case of fMRI recordings of these participants. This, in turn, allows for realization of the involvement of subcortical regions in the observed changes in the pattern of cortical flow of information. The importance of the study of the effect of stress on the flow of information between these cortico-subcortical regions becomes apparent, considering the crucial role of these subcortical regions as the root of emotional responses [5].

While selecting the participants for the present study, the limited number of female participants in the final HIGH and LOW stress-susceptible groups left us with no choice but to exclude them from our further analyses. On the other hand, Lighthall et al. [102] and Seo et al. [72] showed the differential effect of participants’ gender on the brain response to stress. This necessitates further investigation of our findings in settings in which both male and female (as well as more gender-diverse individuals) are included. Similarly, the limited number of older adults (i.e., one individual in each of LOW and HIGH stress-susceptible groups) required us to exclude them from further analyses since their limited sample would have not allowed us to verify whether the observed results were due to stress or the effect of ageing on reduced complexity and information processing capacity of the brain [103,104,105,106,107]. As a result, it is crucial to consider the sample population that comprise older people as well as adolescent [10,27] for drawing more informed conclusion on the change in cortical information processing in response to stress.

It is also important to note that our findings do not readily generalize to the case of overall brain responses to stress. This is because our approach to selecting the individuals for the present study inevitably discarded those participants whose responses fell between the HIGH and LOW stress-susceptible groups. As a result, we were not able to verify whether the observed changes in the cortical flow of information was due to the significantly different mindset of HIGH versus LOW groups (i.e., as far as their subjective responses to the questionnaires were concerned) or it rather captured a substantial change in the brain functions whose gradual effect could be traced along the stress-effect spectrum. Considering the findings that identified the individuals’ subjective ratings to best predict their distress across a variety of self-report measures [108], future research can broaden the scope of the present study to the case in which individuals with broader stress responses are included. This, in turn, can allow for more informed conclusion on generalizability of observed differences between HIGH and LOW stress-susceptible groups to more inclusive scenario in which wider range of stress responses are considered.

The present results suggested TE as a useful tool for the study and analysis of the effect of stress on brain’s distributed cortical information processing. It is apparent that, compared to more conventional correlation/coactivation-based approaches, the use of such measures of directed flow of information as TE can provide a more comprehensive view of the brain’s regional interactivity and information-sharing. Despite this interesting feature, TE suffers from one key criterion, i.e., its expensive computation. Specifically, TE’s time complexity is O(R^3^), where R refers to its computational resolution. In this respect, recent research presented substantial progress on addressing this limitation [109,110]. However, further analytical studies are necessary to allow for harnessing the full utility of TE for quantification of the effect of various psychological and mental disorders on brain function. This is in particular crucial to enable the use of TE as a useful feature for real-time data-driven approaches to decoding of the brain activity [57,58,59,60].

## Figures and Tables

**Figure 1 biology-09-00236-f001:**
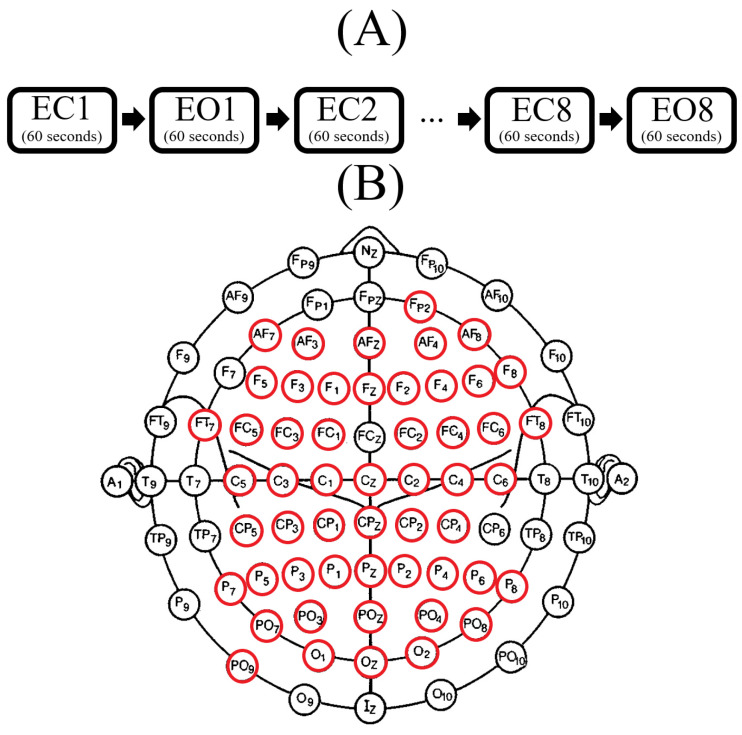
(**A**) Resting-state EEG acquisition protocol in mind–body–brain dataset [53]. It comprised 16 blocks where these blocks (60-s each) were divided into two settings: Eyes-Closed (EC) and Eyes-Open (EO). Each setting consisted of 8 blocks. These EC and EO blocks were interleaved and the recording started with EC for all participants. In this figure, the numbers 1 through 8 refer to the corresponding EC/EO block number. (**B**) Fifty-three EEG channels (circled in red) that were commonly available in all participants’ preprocessed EEG recordings. These channels were FP2, AF7, AF3, AFZ,, AF4, AF8, C5, C3, C1, Cz, C2, C4, C6, CP5, CP3, CP1, CPZ, CP2, CP4, F5, F3, FZ, F1, F2, F4, F6, F8, FT7, FC5, FC3, FC1, FC2, FC4, FC6, FT8, P7, P5, P3, P1, PZ, P2, P4, P6, P8, PO9, PO7, POZ, PO3, PO4, PO8, and O1, OZ, O2.

**Figure 2 biology-09-00236-f002:**
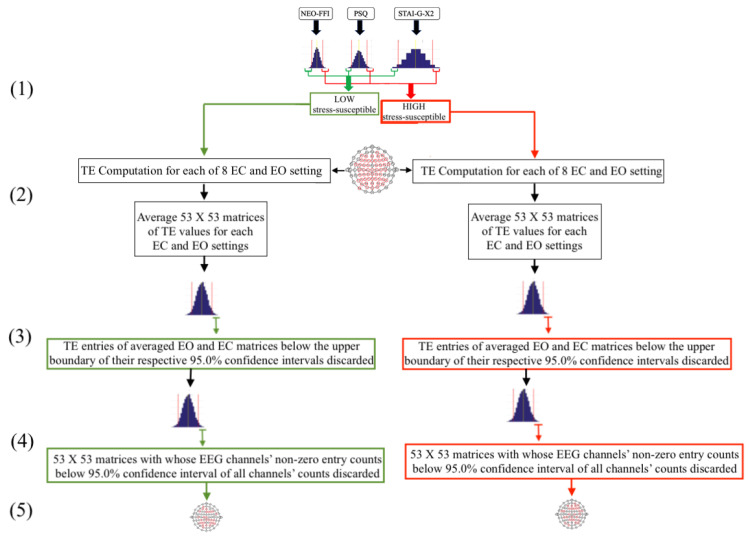
An overview of data inclusion in the present study. The procedure comprised five steps. (1) 95.0% confidence intervals of participants’ responses to each of NEO-FFI, PSQ, and STAI-G-X2 were calculated separately. Subsequently, participants whose responses to each of three questionnaires fell in lower/upper boundary of respective confidence interval of its related questionnaire (indicated by green/red lines, respectively) were placed in LOW/HIGH stress-susceptible groups. (2) Pair-wise TE values for all available 53-EEG-channels (Figure 1B) for each of these LOW and HIGH stress-susceptible participants, per individual, for each of their EC and EO settings’ matrices (i.e., 8 matrices, per setting) were calculated. This resulted in 8 TE matrices of size 53 × 53, per EC and EO settings. This was followed by averaging 8 TE matrices for each of EC and EO settings, per participant, thereby obtaining one averaged 53 × 53 TE matrix, per EC and EO settings, per participant. (3) 95.0% confidence interval for TE values, per EC and EO settings, of all participants in LOW and HIGH stress-susceptible groups were separately computed (i.e., 10 × 53 × 53 = 28090 TE values, per EC and EO settings) and TE values, per participant, per setting, that were below the upper boundary of their respective group’s confidence interval (i.e., per EC and EO settings) were discarded. (4) 95.0% confidence intervals for combined individuals’ number of non-zero entries in each row of the averaged TE matrices, per EC and EO settings, per stress-susceptible group, were computed and EEG channels whose number of non-zero TE entries were below the upper boundary of their related confidence interval were discarded, per individual, per stress-susceptible groups, and per EC and EO settings. (5) Union of EEG channels among individuals in each of LOW and HIGH stress-susceptible groups, per EC and EO settings, were separately selected (i.e., two EEG channels’ arrangements at the bottom, Figure A2). In the case of EC, we found 18 EEG channels that survived these thresholding steps and that were common between all participants in HIGH stress-susceptible group (AFZ, AF4, F1, F6, FT7, CZ, C2, C6, CP5, CP3, CP1, CP4, P7, P5, P4, P6, PO3, POZ). We also found 18 surviving EEG channels that were common among all participants in LOW stress-susceptible group (FP2, AFZ, F3, F5, FZ, CP1, CP3, CP5, CPZ, P7, P3, P4, P2, P6, PO3, POZ, PO4, O2). We used the union (without repetition) of these EEG channels for comparative analyses between HIGH and LOW stress-susceptible groups during EC setting (Figure A2A). In the case of EO, we found 21 EEG channels that survived these thresholding steps and that were common between all participants in HIGH stress-susceptible group (AF3, AF4, C1, C2, C6, CP3, CP4, CP5, F1, F2, F6, FC5, FC6, FT7, FT8, Fp2, Fz, O1, P1, P6, PO4). The number of such EEG channels in LOW stress-susceptible group was 17 (AF3, AF8, AFZ, C2, CP3, CPZ, F2, F4, FC5, FC6, FT7, Fp2, Fz, P2, PO7, PO8, PZ). Similar to the case of EC, we used the union of these EEG channels for comparative analyses between HIGH and LOW stress-susceptible groups for EO setting (Figure A2B).

**Figure 3 biology-09-00236-f003:**
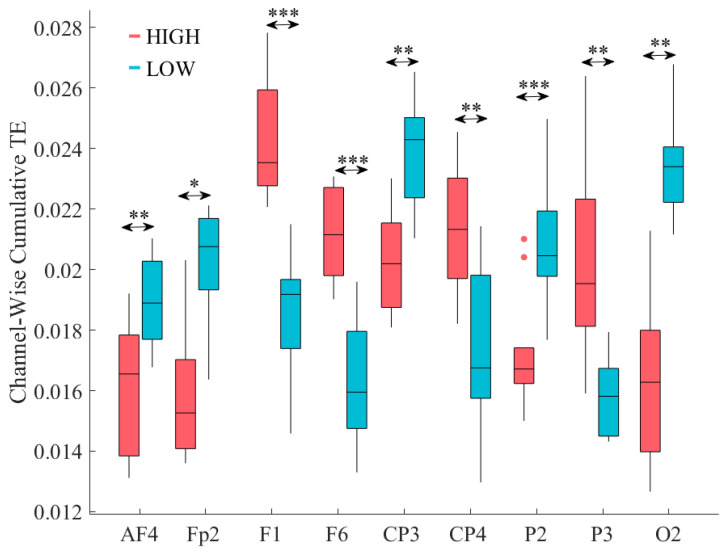
Eyes-Closed (EC) setting. Channel-wise paired Wilcoxon rank sum test between total TEs of participants in HIGH and LOW stress-susceptible groups. There were significant differences between total TEs of nine EEG channels. They were: FP2, AF4, F1, F6, CP3, CP4, P2, P3, and O2. In this figure, the asterisks mark these significant differences (* *p* < 0.05, ** *p* < 0.01, *** *p* < 0.001).

**Figure 4 biology-09-00236-f004:**
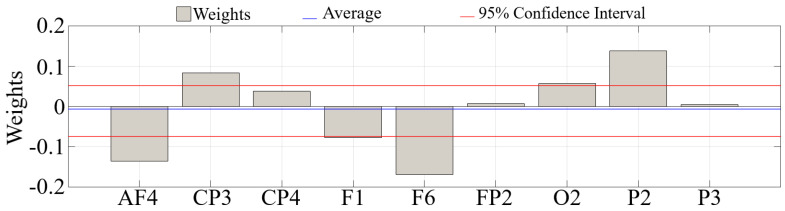
Eyes-Closed (EC) GLM analysis using logistic regression with sigmoid function to determine the importance of each of the significantly different channels between HIGH and LOW stress-susceptible groups for predicting participants’ group membership (i.e., HIGH versus LOW). Channels with significantly different total TEs were FP2, AF4, F1, F6, CP3, CP4, P2, P3, and O2. ANOVA analysis of these weights indicated non-significant difference between contribution of these channels individually and in comparison to other channels’ contribution. Blue line in each subplot marks the average of these coefficients, per setting. Red lines mark the 95.0% confidence interval of these weights.

**Figure 5 biology-09-00236-f005:**
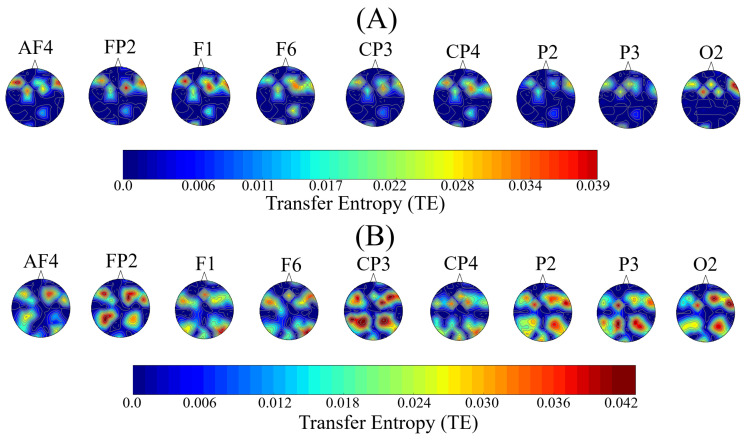
Eyes-Closed (EC) setting. Global distribution of TEs of channels whose total TE values significantly differed between HIGH and LOW stress-susceptible groups. These channels were FP2, AF4, F1, F6, CP3, CP4, P2, P3, and O2. (**A**) HIGH stress-susceptible group (**B**) LOW stress-susceptible group. This figure indicates that distributed TEs from these channels were primarily contributed to the frontal regions in the case of HIGH stress-susceptible group. On the other hand, their contribution was more globally distributed among fronto-parietal in the case of LOW stress-susceptible group.

**Figure 6 biology-09-00236-f006:**
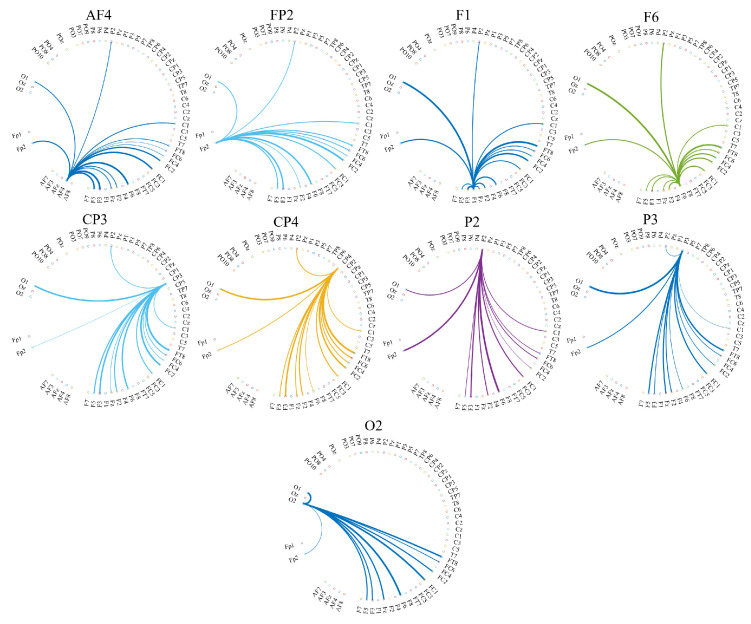
Eyes-Closed (EC) setting. Transfer of information from FP2, AF4, F1, F6, CP3, CP4, P2, P3, and O2 to the other cortical regions in the case of HIGH stress-susceptible group. In each subplot, the source of information transfer (e.g., AF4, FP2, etc.) is the one from which all connections are originated. They are labeled at the top of each subplot. The recipients of information (i.e., channels on the other cortical regions) are the ones at the other end of these outreaching arches.

**Figure 7 biology-09-00236-f007:**
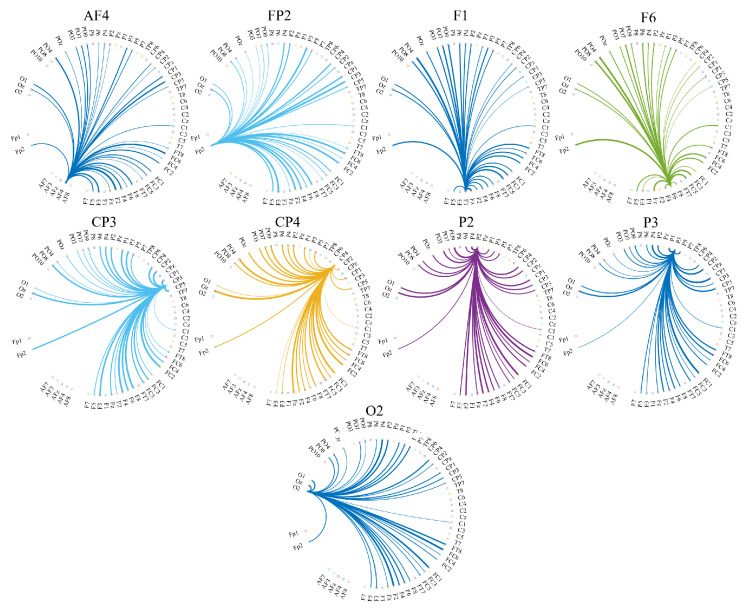
Eyes-Closed (EC) setting. Transfer of information from (i.e., out-degree) FP2, AF4, F1, F6, CP3, CP4, P2, P3, and O2 to the other cortical regions in the case of LOW stress-susceptible group. In each subplot, the source of information transfer (e.g., FP2, AF4, etc.) is the one from which all connections are originated. They are labeled at the top of each subplot. The recipients of information (i.e., channels on the other cortical regions) are the ones at the other end of these outreaching arches.

**Table 1 biology-09-00236-t001:** Eyes-Closed (EC) setting. Channel-wise paired Wilcoxon rank sum test for four channels (AF4, F1, F6, and CP4) in which significantly higher total TEs were associated with HIGH stress-susceptible group. *p*, *W*, and *r* refer to *p*-value, test-statistics, and the effect size for Wilcoxon rank sum tests. M and SD are the mean and standard deviation of these channels’ total TE values. The subscripts to M and SD mark the two groups. All these significant differences were characterized by strong effect sizes, as indicated by the “*r*” column entry of this table.

Channel	*p* =	*W*(18)	*r*	$M_{\mathrm{HIGH}}$	${\mathrm{SD}}_{\mathrm{HIGH}}$	$M_{\mathrm{LOW}}$	${\mathrm{SD}}_{\mathrm{LOW}}$
AF4	0.0010	3.29	0.74	0.020	0.003	0.016	0.001
F1	0.0002	3.74	0.84	0.024	0.002	0.019	0.002
F6	0.0004	3.52	0.79	0.021	0.001	0.016	0.002
CP4	0.0091	2.61	0.58	0.021	0.002	0.017	0.003

**Table 2 biology-09-00236-t002:** Eyes-Closed (EC) setting. Channel-wise paired Wilcoxon rank sum test for five channels (FP2, CP3, P2, P3, and O2) in which higher total TEs were associated with LOW stress-susceptible group. *p*, *W*, and *r* refer to *p*-value, test-statistics, and the effect size for Wilcoxon rank sum tests. M and SD are the mean and standard deviation of these channels’ total TE values. The subscripts to M and SD mark the two groups. All these significant differences were characterized by strong effect sizes, as indicated by the “*r*” column entry of this table.

Channel	*p* =	*W*(18)	*r*	$M_{\mathrm{HIGH}}$	${\mathrm{SD}}_{\mathrm{HIGH}}$	$M_{\mathrm{LOW}}$	${\mathrm{SD}}_{\mathrm{LOW}}$
FP2	0.0100	−2.47	0.55	0.016	0.002	0.0190	0.002
CP3	0.0013	−3.21	0.72	0.020	0.002	0.024	0.002
P2	0.0003	−3.59	0.80	0.016	0.003	0.023	0.002
P3	0.0036	−2.91	0.65	0.017	0.002	0.021	0.002
O2	0.0013	−3.21	0.72	0.016	0.0021	0.020	0.002

**Table 3 biology-09-00236-t003:** Eyes-Closed (EC) setting. ANOVA analysis of GLM coefficients associated with the channels with significant difference in their total TEs between HIGH and LOW stress-susceptible groups. These channels were: FP2, AF4, F1, F6, CP3, CP4, P2, P3, and O2. This analysis identified a non-significant difference between contribution of these channels individually and in comparison with the other channels’ contribution.

Channel	*p* =	F	$\eta^{2}$
AF4	0.6734	0.19	0.02
FP2	0.9813	F = 0.001	0.0001
F1	0.8818	F = 0.02	0.002
F6	0.6864	0.17	0.014
CP3	0.8292	0.05	0.004
CP4	0.9290	0.008	0.0007
P2	0.8000	0.07	0.006
P3	0.9893	F = 0.0002	0.00002
O2	0.8915	0.020	0.002

**Table 4 biology-09-00236-t004:** Eyes-Closed (EC) setting. Channel-wise paired Wilcoxon rank sum tests for four channels (AF4, F1, F6, and CP4) in which significantly higher distributed TEs were associated with HIGH stress-susceptible group. *p*, *W*, and *r* refer to *p*-value, test-statistics, and the effect size for Wilcoxon rank sum tests. M and SD are the mean and standard deviation of these channels’ distributed TE values for HIGH and LOW stress-susceptible groups. The subscripts to M and SD mark the two groups.

Channel	*p* =	*W*(1058)	*r*	$M_{\mathrm{HIGH}}$	${\mathrm{SD}}_{\mathrm{HIGH}}$	$M_{\mathrm{LOW}}$	${\mathrm{SD}}_{\mathrm{LOW}}$
AF4	0.0003	3.61	0.11	0.020	0.018	0.016	0.018
F1	0.000001	4.50	0.15	0.024	0.018	0.019	0.018
F6	0.00003	4.18	0.13	0.021	0.018	0.016	0.018
CP4	0.00003	4.20	0.13	0.021	0.018	0.017	0.018

**Table 5 biology-09-00236-t005:** Eyes-Closed (EC) setting. Channel-wise paired Wilcoxon rank sum tests for five channels (FP2, CP3, P2, P3, and O2) in which significantly higher distributed TEs were associated with LOW stress-susceptible group. *p*, *W*, and *r* refer to *p*-value, test-statistics, and the effect size for Wilcoxon rank sum tests. M and SD are the mean and standard deviation of these channels’ distributed TE values for HIGH and LOW stress-susceptible groups. The subscripts to M and SD mark the two groups.

Channel	*p* =	*W*(1058)	*r*	$M_{\mathrm{HIGH}}$	${\mathrm{SD}}_{\mathrm{HIGH}}$	$M_{\mathrm{LOW}}$	${\mathrm{SD}}_{\mathrm{LOW}}$
FP2	0.0052	−2.79	0.09	0.016	0.018	0.019	0.019
CP3	0.0002	−3.71	0.11	0.020	0.019	0.024	0.018
P2	0.0000	5.46	0.17	0.016	0.018	0.023	0.018
P3	0.0014	−3.19	0.10	0.017	0.019	0.021	0.018
O2	0.0009	−3.33	0.10	0.016	0.019	0.020	0.019

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
