# Peer review of "Stress Changes the Resting-State Cortical Flow of Information from Distributed to Frontally Directed Patterns"

_biology, 2020, doi:10.3390/biology9080236_

Round 1
Reviewer 1 Report
The authors presented an interesting study attempting to find functional connectivity patterns related to the stress levels.
They analyzed EEG resting-state recordings from two conditions: eyes-open and eyes-closed.
The study is interesting but there are many issues that should be addressed properly.
Below, I summarize my comments.
1) The authors analyzed EEG recording at broadband activity using a bandpass Butterworth filtering of eight order.
Does this mean that you applied a filter of four-order of zero-phase (in both directions.)?
2) You reported that you denoised EEG multi-channel recordings using ICA
but you didn't report:
a) which software did you use and with which settings?
b) how did you detect artifactual IC?
c) how many artifactual ICs did you detect on average per condition?
d) did you concatenate the recordings across the cohort before applying the ICA?
3) You estimated the information flow between every pair of EEG using TE estimating also the time-lag via the maximization of TE.
Section 2.5.3 reports the adopted thresholding criterion but I cannot understand what did you really estimate and how.
You have to explain it in more detail.
How does bootstrap was estimated in both cases of threshold and the counting of non-zeros?
What does the following statement mean?
the count of non-zero TE entries of each EEG channel (i.e., the channels that
each channel transferred information to)
TE gives you the possibility of finding the sender and the receiver.
No-zero TE entries refer to TEs across pairs of channels with a weight survived the bootstrapping?
4) One big mistake in the paper is that you reported your findings in the methods part.
Methods part should describe only the methodology part without giving
any research finding.
5) You reported :
We found 18 EEG channels that survived these thresholding steps and that were common between all participants in HIGH stress-susceptible group.
A)Does this mean that your bootstrapping analysis has been performed per subject and the your findings were consistent across the cohort?
B) 18 EEG channels were detected from the connected sub-networks with an average of 26.65 connections in the high group?
You have to connect this report with tables 4 & 5.
6) In section 2.6.2, you reported:
For each of the selected channels for HIGH and LOW stress-susceptible groups (Figure 3) in both EC and EO settings, we first computed its total TE.We calculated the total TE as the cumulative sum of information that was transferred by a channel (e.g., AF4) to the remaining fifty-two EEG channels
(Figure 2 (B)).
You practically estimated the strength of every sensor.
A) You have to report that you estimated the functional strength of the 27
EEG sensors and not from the original set of EEG sensors.
The union process of EEG channels from both conditions define a subset of 27 EEG sensors.
B) Did you report the out-strength or the in-strength or both?
then, you reported:
We then performed Wilcoxon rank sum test between each pair of selected channels for HIGH and LOW stress-susceptible groups (e.g., AF4 in HIGH and LOW). We carried out this analysis on EEG channels of both, EC (Figure 3 (A)) and EO (Figure 3 (B)) settings. These tests identified a number of EEG channels that showed significantly different total TEs between HIGH and LOW stress-susceptible participants.
C) This approach is a supervised procedure that guides the process of discriminating between the two groups. For that reason, the following described RSA analysis is of no importance since you directed the feature a prior by using the labels of low/high.
7) In section 2.6.3, you reported again the estimation of the total strength
between the selected EEG sensors between low-high group in both conditions with the rest of the network.
However, this is completely out of research strategy based on your approach.
In previous sections, you selected the union of EEG sensors from both conditions with non-zero entries.
This means that only 18 - 21 EEG sensors in both conditions are connected via TE.
This section introduces a strong bias and a big misunderstanding to the whole process.
8) In figure 7, did you report out-strength , in-strength or the whole-strength.
This should be reported clearly across the whole study.
Which is of three types of strength did you use even in your glm analysis?
Author Response
First and foremost, the author would like to take this opportunity to express his gratitude for the reviewer’s time and kind consideration to review the present manuscript. The reviewer’s comments substantially improved the quality of the present study and its presentation.
In what follows, point-by-point responses to the reviewer’s comments and concerns are provided.
Sincerely,
Reviewer 1
Prior to responding to the reviewer’s comments, the author would like to bring to reviewer’s kind attention that the present study used the pre-processed EEG data that were provided by the Max Planck Institute Leipzig Mind-Brain-Body Dataset [53]. This choice was to increase the reproducibility of the results presented in the present manuscript. Upon receiving the comments by the reviewer and in order to provide further information on reported pre-processing steps in [53], author found another publication in PLOS Computational Biology that was coauthored by one of the authors in [53] and used the same EEG dataset (reference [70] in the present version of manuscript). Therefore, the author of the present manuscript used this latter publication to fill in some of the missing information with regards to pre-processing steps. Subsequently, reference [70] is cited accordingly and as appropriate.
Reviewer’s Comment: 1) The authors analyzed EEG recording at broadband activity using a bandpass Butterworth filtering of eight order.
Does this mean that you applied a filter of four-order of zero-phase (in both directions.)?
Author’s Response: Although this information was not reported in [53], the author of the present manuscript found it in [70] which reported four-order in both directions to minimize zero-crossing by low-frequency drifts. This information is added to the current version of the manuscript as follows (Section 2.3. Preprocessing, lines 136-138).
“… It was then, per channel, per participant, bandpass-filtered within 1-45 Hz using an eight-order Butterworth filter (i.e., four-order in both directions to minimize zero-crossing distortions by low-frequency drifts [70]).”
Reviewer’s Comment: 2) You reported that you denoised EEG multi-channel recordings using ICA
but you didn't report:
- a) which software did you use and with which settings?
Author’s Response: With regard to ICA, [53] reported that Infomax (runica) algorithm was used. This information is added to Section 2.3. Preprocessing, lines 144-146, in the current version of manuscript. It reads as follows.
“This step was then followed by independent component analysis (ICA) on temporal (i.e., EEG channels' data points) dimension of data using the Infomax (runica) algorithm.”
[53] also reported that their analyses of EEG data were performed in EEGLAB (version 14.1.1b) for MATLAB (Delorme and Makeig, 2004). This information is added to Section 2.3. Preprocessing, lines 151-152, in the current version of manuscript. It reads as follows.
“These analyses were performed using EEGLAB [71] (version 14.1.1b) for MATLAB (Delorme and Makeig, 2004).”
- b) how did you detect artifactual IC?
Author’s Response: [53] reported that the components that reflected eye movement, eye blink, or heartbeat related artifacts were removed and that the retained independent components for EO (M = 19.70, range = 9.0-30.0) and EC (M = 21.40, range = 14.0-28.0) conditions were back-projected to the sensor space for further analysis. In addition, [70] reported the following settings for ICA: step size: , annealing policy: when weight change > 0.000001, learning rate is multiplied by 0.98, stopping criterion maximum number of iterations 512 or weight change < 0.000001. This information is added to Section 2.3. Preprocessing, lines 144-148, in the current version of manuscript. It reads as follows.
“This step was then followed by independent component analysis (ICA) on temporal (i.e., EEG channels' data points) dimension of data using the Infomax (runica) algorithm (step size: , annealing policy: when weight change > 0.000001, learning rate was multiplied by 0.98, stopping criterion maximum number of iterations 512 or weight change < 0.000001).”
- c) how many artifactual ICs did you detect on average per condition?
Author’s Response: [53] reported 21.40, range = 9.0-30.0. This information is added to Section 2.3. Preprocessing, lines 148-151, in the current version of manuscript. It reads as follows.
“Subsequently, components that reflected eye movement, eye blink, or heartbeat related artifacts were removed. Retained independent components for EO (mean number of component = 21.40, range = 9.0-30.0) and EC (M = 21.40, range = 14.0-28.0) conditions were then back-projected to the sensor space for further analysis.”
- d) did you concatenate the recordings across the cohort before applying the ICA?
Author’s Response: [53] reported that the raw EEG data (after downsampling, followed by bandpass filtering), were split into EO and EC conditions for the subsequent analyses. They also reported that (1) outlier channels were rejected after visual inspection for frequent jumps/shifts in voltage and poor signal quality (2) data intervals containing extreme peak-to-peak deflections or large bursts of high frequency activity were identified by visual inspection and removed (3)intervals containing traces from eye blinks or eye movements were not removed at this stage. These three steps were then followed by such steps as applying PCA and ICA, etc. Apart from these information, no further information was provided in [53] (this was also the case in [70]). The highlighted information in this response are added to Section 2.3. Preprocessing, lines 138-142, in the current version of manuscript as follows.
“The data was then split into EC and EO, each comprising eight 60-second blocks. For each of these blocks, the EEG channels were visually inspected and the channels that were affected by such issues as frequent jumps/shifts in voltage and/or poor signal quality were rejected. During this step, data intervals that contained extreme peak-to-peak deflections or large bursts of high frequency activity were also removed (identified through visual inspection).”
Reviewer’s Comment: 3) You estimated the information flow between every pair of EEG using TE estimating also the time-lag via the maximization of TE.
Section 2.5.3 reports the adopted thresholding criterion but I cannot understand what did you really estimate and how.You have to explain it in more detail.
How does bootstrap was estimated in both cases of threshold and the counting of non-zeros?
What does the following statement mean?
the count of non-zero TE entries of each EEG channel (i.e., the channels that
each channel transferred information to)
TE gives you the possibility of finding the sender and the receiver.
No-zero TE entries refer to TEs across pairs of channels with a weight survived the bootstrapping?
Author’s response: The reviewer’s comment is addressed in the following two steps.
- First, we added a new section (Section 2.4. An Overview of the Participants' Selection and EEG Inclusion Process, lines 165-208, in the current version of manuscript) that summarizes the procedure through which the participants and their EEG channels for each of HIGH and LOW stress-susceptible groups, per EC and EO settings, are selected. This Section also includes a new figure (Figure 2, page 6, in the current version of manuscript) that illustrates the steps involved in this process.
- Next, detailed results related to these steps that were previously reported in Section 2. Methods are moved to a new Appendix (Appendix A Detailed Participants and EEG Selection Procedure, lines 513-647, in the current version of manuscript). The Sections (along with their corresponding results) that are moved to this Appendix include:
- Appendix A.1 Determination of Participants’ Stress-Susceptibility (lines 504-526, in the current version of manuscript).
- Appendix 1.1 Participants’ Selection Based on Their Responses to NEO-FFI, PSQ, STAI-G-X2 (lines 539-586, in the current version of manuscript).
- Appendix A.2 Quantification of the Directed Transfer of Information Between EEG Channels (lines 587-592, in the current version of manuscript).
- Appendix A.2.1 Overview of Transfer Entropy (TE) (lines 593-596, in the current version of manuscript).
- Appendix A.2.2 Participants’ TE Computation (lines 599-618, in the current version of manuscript).
- Appendix A.2.3 Thresholding the Participants’ TEs (lines 619-647, in the current version of manuscript).
- With regard to the part of reviewer’s comment “What does the following statement mean?”: the concerned sentence is modified as follows
- In the new Section 2.4. An Overview of the Participants' Selection and EEG Inclusion Process (lines 186-192, in the current version of manuscript):
“In step four (Figure 2 (4)), we first counted the number of non-zero entries in each row of the averaged TE matrices, per individual, per EC and EO settings. Next, we combined the individuals' counts for LOW and HIGH stress-susceptible groups separately and computed the 95.0% confidence intervals for these counts (i.e., per EC and EO settings, per stress-suceptible groups). We then discarded those EEG channels whose number of non-zero TE entries were below the upper boundary of their related confidence interval, per individual, per stress-susceptible groups, and per EC and EO settings.”
- In Appendix A.2.3 Thresholding the Participants’ TEs (lines 626-631):
“Next, we counted the number of non-zero entries in each row of the averaged TE matrices, per individual, per EC (Table A4) and EO (Table A5). Next, we combined the individuals' counts for LOW and HIGH stress-susceptible groups separately and computed the 95.0% confidence intervals for these counts (i.e., per EC and EO settings, per stress-suceptible groups). We then discarded those EEG channels whose number of non-zero TE entries were of their related confidence interval, per individual, per stress-susceptible groups, and per EC and EO settings.”
Reviewer’s Comment: 4) One big mistake in the paper is that you reported your findings in the methods part.
Methods part should describe only the methodology part without giving
any research finding.
Authors’ Response: Please refer to the author’s response to the reviewer’s comment “3) You estimated the information flow between every pair of EEG using…”
Reviewer’s Comment: 5) You reported :
We found 18 EEG channels that survived these thresholding steps and that were common between all participants in HIGH stress-susceptible group.
A)Does this mean that your bootstrapping analysis has been performed per subject and the your findings were consistent across the cohort?
Author’s Response: This step is further clarified in the current version of manuscript (Section 2.4. An Overview of the Participants’ Selection and EEG Inclusion Process, fourth point, lines 186-192 and fifth point, lines 193-208). They read as follow.
- Section 2.4. An Overview of the Participants’ Selection and EEG Inclusion Process, fourth point, lines 186-192:
“we first counted the number of non-zero entries in each row of the averaged TE matrices, per individual, per EC and EO settings. Next, we combined the individuals' counts for LOW and HIGH stress-susceptible groups separately and computed the 95.0% confidence intervals for these counts (i.e., per EC and EO settings, per stress-suceptible groups). We then discarded those EEG channels whose number of non-zero TE entries were below the upper boundary of their related confidence interval, per individual, per stress-susceptible groups, and per EC and EO settings.”
- Section 2.4. An Overview of the Participants’ Selection and EEG Inclusion Process, fifth point, lines 193-208:
“In step five (Figure 2 (5)) we separately found the union of EEG channels among individuals in each of LOW and HIGH stress-susceptible groups, per EC and EO settings (Figure A2). In the case of EC, this step resulted in 18 EEG channels that were common between all participants in HIGH stress-susceptible group. These channels were AFZ, AF4, F1, F6, FT7, CZ, C2, C6, CP5, CP3, CP1, CP4, P7, P5, P4, P6, PO3, POZ. Similarly, there were 18 EEG channels that were common among all participants in LOW stress-susceptible group. They were FP2, AFZ, F3, F5, FZ, CP1, CP3, CP5, CPZ, P7, P3, P4, P2, P6, PO3, POZ, PO4, O2. We used the union of these EEG channels (without repetition) for comparative analyses between HIGH and LOW stress-susceptible groups in EC setting (Figure A2 (A)). On the other hand, we found 21 EEG channels in EO setting that survived these thresholding steps and that were common between all participants in HIGH stress-susceptible group. Those channels were AF3, AF4, C1, C2, C6, CP3, CP4, CP5, F1, F2, F6, FC5, FC6, FT7, FT8, Fp2, Fz, O1, P1, P6, PO4. The number of such EEG channels in LOW stress-susceptible group was 17 (AF3, AF8, AFZ, C2, CP3, CPZ, F2, F4, FC5, FC6, FT7, Fp2, Fz, P2, PO7, PO8, PZ). Similar to the case of EC, we used the union of these EEG channels for comparative analyses between HIGH and LOW stress-susceptible groups for EO setting (Figure A2(B)).”
- B) 18 EEG channels were detected from the connected sub-networks with an average of 26.65 connections in the high group?
You have to connect this report with tables 4 & 5.
Author’s Response: This part is modified to more clearly explain the step involved and connect it with Tables 4 & 5 (i.e., Tables A4 and A5, in current version of manuscript). Its content reads as follows (Appendix A.2.3 Thresholding the Participants’ TEs (lines 626-631)).
“Next, we counted the number of non-zero entries in each row of the averaged TE matrices, per individual, per EC (TableA4) and EO (Table A5). Next, we combined the individuals' counts for LOW and HIGH stress-susceptible groups separately and computed the 95.0% confidence intervals for these counts (i.e., per EC and EO settings, per stress-suceptible groups). We then discarded those EEG channels whose number of non-zero TE entries were of their related confidence interval, per individual, per stress-susceptible groups, and per EC and EO settings.”
Reviewer’s Comment: 6) In section 2.6.2, you reported:
For each of the selected channels for HIGH and LOW stress-susceptible groups (Figure 3) in both EC and EO settings, we first computed its total TE.We calculated the total TE as the cumulative sum of information that was transferred by a channel (e.g., AF4) to the remaining fifty-two EEG channelsn (Figure 2 (B)).
You practically estimated the strength of every sensor.
- A) You have to report that you estimated the functional strength of the 27 EEG sensors and not from the original set of EEG sensors.
The union process of EEG channels from both conditions define a subset of 27 EEG sensors.
Author’s Response: To more clearly represent the total TE computation, a new equation (equation (1), page 8, in the current version of manuscript) is added. Furthermore, the content of this paragraph is also modified to reflect the note by the reviewer and also to increase its readability (Section 2.5.2 Total TEs, lines 240-250, in the current version of manuscript). The author apologizes for not including the modified content here. This is due to the presence of the new equation and some additional symbols that may lose their formatting if incorporated in this review response.
- B) Did you report the out-strength or the in-strength or both?
Author’s Response: Total TEs are computed by summing the row entries of TE matrices (i.e., one row, per EEG channel, Section 2.5.2 Total TEs, lines 240-250, in the current version of manuscript), for each participant and in each EC and EO settings. Therefore, it is indeed the total information transferred by every channel (i.e., out-strength) to other channels. To further clarify this point, the following sentence is added to Section 2.5.2 Total TEs, lines 248-250.
“It is also worthy of note that since total TEs were computed by summing the row entries of the participants' TE matrices, they quantified each EEG channels' functional strength in terms of their total information transferred to every other EEG channel.”
then, you reported:
We then performed Wilcoxon rank sum test between each pair of selected channels for HIGH and LOW stress-susceptible groups (e.g., AF4 in HIGH and LOW). We carried out this analysis on EEG channels of both, EC (Figure 3 (A)) and EO (Figure 3 (B)) settings. These tests identified a number of EEG channels that showed significantly different total TEs between HIGH and LOW stress-susceptible participants.
- C) This approach is a supervised procedure that guides the process of discriminating between the two groups. For that reason, the following described RSA analysis is of no importance since you directed the feature a prior by using the labels of low/high.
Author’s Response: Wilcoxon test is a non-parametric test of significance that tests the null hypothesis that data in two sets X and Y are samples from continuous distributions with equal medians, against the alternative that they are not. The test assumes that the two samples are independent. Considering it as a supervised procedure is synonymous to interpreting any statistical test (whether parametric such as t-test or non-parametric) as supervised since they are all meant to verify whether two sets of values (e.g., X and Y above) are (non-)significantly different from each other. Moreover, applying this test on two data from HIGH and LOW stress-susceptible groups does not have any effect on RSA for the following reasons.
- First, it is possible to have two sets of values whose difference is non-significant and still perfectly distinguishable from each other. For example, the following two hypothetical concentric sets (please refer to the PDF version of author's responses for the associated figure)
X: red
Y = blue
It is apparent that and since they are concentric, their mean and median coincide (i.e., the value at their coincidental center). As a result, both, testing for their difference based on mean (e.g., t-test) or median (e.g., Wilcoxon) will yield a non-significant difference. However, X and Y can be perfectly distinguished from each other by any simple linear model with polynomial terms (in fact, their decision boundary lies at + where “r” is the radius of X).
On the other hand, the setting such as the figure below (please refer to the PDF version of author's responses for the associated figure) results in a significant difference between X and Y without any trivial boundary to distinguish them with high accuracy.
- RSA and Wilcoxon test are two disjoint results whose performances are not derived from one another. More specifically, RSA could have been applied without requiring to perform Wilcoxon test and vise versa. Their results indeed highlight two distinct properties of HIGH and LOW stress-susceptible groups’ TE values. Specifically, Wilcoxon test identified the flow of information between cortical regions in these two different groups were significantly different. On the other hand, RSA indicated that their differences resulted in highly different cortical flow of information that distinguished them with high accuracy.
However, the author realizes that the presentation of this part of manuscript was not properly done and could have therefore led the readers to misinterpretation. Therefore, its content was modified as follows (lines 260-281) as follows.
“We used these EEG channels, per EC and EO settings, and performed two additional analyses on them, thereby evaluating their specificity and sensitivity to differentiate between HIGH and LOW stress-susceptible groups. These analyses were
- Representational Similarity Analysis (RSA) [73,74]: we applied RSA with Euclidean similarity distance on these channels. The input to RSA was the vectors of significantly different EEG channels of the participants (i.e., 1 X 9 and 1 X 14 vectors, per participant, for each of the EC and EO settings). We then performed Wilcoxon rank sum tests on HIGH and LOW RSA-based clusters (both within- and between-cluster). Next, we adapted a one-holdout cross-validation strategy to determine how well the individuals from each of these two groups could be predicted. For this purpose, we separated one of the participants from HIGH/LOW stress-susceptible group and computed the Euclidean similarity distances between all the remaining participants. We then predicted the group membership of the holdout participant by computing the vicinity of this participant’s vector to the center of HIGH and LOW RSA-based clusters. We repeated this step for every individual (i.e., 10 HIGH and 10 LOW participants), per EC and EO settings.
- General Linear Model (GLM) [75,76]: To realize the importance of each of these EEG channels, we opted for GLM analysis using logistic regression with sigmoid function. The inputs to this model were N X M matrices where N refers to the number of participants (i.e., 20 in our case) and M is the number of channels whose total TEs significantly differed between HIGH and LOW groups (i.e., 9 and 14 in the case of EC and EO, respectively). We used 1 and 0 as class labels for HIGH and LOW stress-susceptible groups, per EC and EO settings. We then carried out ANOVA analyses on this model’s coefficients (i.e., weights), per EC and EO settings. We used Matlab 2016a ("fitlm" and its corresponding "anova" functions) for these analyses”
Reviewer’s Comment: 7) In section 2.6.3, you reported again the estimation of the total strength
between the selected EEG sensors between low-high group in both conditions with the rest of the network.
However, this is completely out of research strategy based on your approach.
In previous sections, you selected the union of EEG sensors from both conditions with non-zero entries.
This means that only 18 - 21 EEG sensors in both conditions are connected via TE.
This section introduces a strong bias and a big misunderstanding to the whole process.
Author’s Response: This Section (Section 2.5.3 Distributed TEs, lines 284-297) is not out of research strategy but a complementary step that pinpoint how the observed changes based on total TE is indeed distributed among cortical regions. Total TE is nothing but the sum of these distributed TEs (Section 2.5.2 Total TEs, lines 240-250). As a result, this part of analyses identifies how each channel’s distributed transfer of information to other cortical regions was affected by stress-susceptibility (i.e., LOW vs. HIGH groups).
Reviewer’s Comment: 8) In figure 7, did you report out-strength , in-strength or the whole-strength.
This should be reported clearly across the whole study.
Author’s Response: The caption of this figure is modified by adding “(i.e., out-degree)” after “from.” It reads as follows.
“Eyes-Close (EC) setting. Transfer of information from (i.e., out-degree) FP2, AF4, F1, F6, CP3, CP4, P2, P3, and O2 to the other cortical regions in the case of LOW stress-susceptible group. In each subplot, the source of information transfer (e.g., FP2, AF4, etc.) is the one from which all connections are originated. They are labeled at the top of each subplot. The recipients of information (i.e., channels on the other cortical regions) are the ones at the other end of these outreaching arches.”
Similar change is applied to Figures 8, A7, and A8.
Author’s Response: Which is of three types of strength did you use even in your glm analysis?
Author’s Response: please refer to the author’s response to the reviewer’s comment “6) In section 2.6.2, you reported:… B) Did you report the out-strength or the in-strength or both?”
With regard to GLM, it was based on total TE values (please refer to the author’s response to the reviewer’s comment “C) This approach is a supervised procedure that guides the process of discriminating between”).

Reviewer 2 Report
The paper discusses an important subject. The advances in the data-acquiring process as well as the large extend of systems necessitates having efficient tools, such as the one proposed in this draft. The paper is well written and organized. My comments are as follows:
-While the abstract has aimed to provide an overall overview about the main contribution, there is a need to be revised in such a way that the general reader can grasp the main idea/topic of the draft as well as the main contribution.
-Despite the fact that a good discussion about the superiority of the proposed framework in provided in terms of the numerical results, discussion about the complexity of the proposed framework and how it compares with the existing techniques are highly recommended.
-Having a nice schematic diagram in the draft would be really helpful. This definitely alleviates the difficulty of going to details of the techniques for the readers.
- There have been a surge in the application of Machine Learning and Statistical framework to solve the similar problem focused in this paper. The authors are encouraged to include some of the recent papers in the introduction to give a good holistic overview about the existing techniques to general readers:
# "Estimating regional cerebral blood flow using resting-state functional MRI via machine learning." Journal of neuroscience methods 331 (2020): 108528.
# "Gene regulatory network state estimation from arbitrary correlated measurements." EURASIP Journal on Advances in Signal Processing 2018, no. 1 (2018): 1-10.
# "Machine learning: assessing neurovascular signals in the prefrontal cortex with non-invasive bimodal electro-optical neuroimaging in opiate addiction." Scientific reports 9, no. 1 (2019): 1-14.
- The format of some the references is not in standard form. These need to be checked and fixed.
Author Response
First and foremost, the author would like to take this opportunity to express his gratitude for the reviewer’s time and kind consideration to review the present manuscript. The reviewer’s comments substantially improved the quality of the present study and its presentation.
In what follows, point-by-point responses to the reviewer’s comments and concerns are provided.
Sincerely,
Reviewer 2
Reviewer’s Comment: -While the abstract has aimed to provide an overall overview about the main contribution, there is a need to be revised in such a way that the general reader can grasp the main idea/topic of the draft as well as the main contribution.
Author’s Response: Abstract is modified as follows (lines 14-26, in the current version of the manuscript).
“The contribution of the present study is threefold. First, it identifies that the stress-susceptibility is charactrized by the change in flow of information in fronto-parietal brain network. Second, it shows that these regions are distributed bi-hemispherically and are sufficient to significantly differentiate between the individuals with high versus low stress-susceptibility. Third, it verifies that the high stress-susceptibility is markedly associated with a higher parietal-to-frontal flow of information. These results provide further evidence for the viewpoint in which the brain’s modulation of information is not necessarily accompanied by the change in its regional activity. They further construe the effect of stress in terms of a disturbance that disrupts the flow of information among the brain's distributed cortical regions. These observations, in turn, suggest that some of the differences in the previous findings perhaps reflect different aspects of impaired distributed brain information processing in response to stress. From a broader perspective, these results posit the use of TE as a potential diagnostic/prognostic tool in identification of the effect of stress on distributed brain networks that are involved in stress-response.”
Reviewer’s Comment: -Despite the fact that a good discussion about the superiority of the proposed framework in provided in terms of the numerical results, discussion about the complexity of the proposed framework and how it compares with the existing techniques are highly recommended.
Author’s Response: The following paragraph is added to Section Limitations and Future Direction, lines 494-504, in the current version of manuscript.
“The present results suggested TE as a useful tool for the study and analysis of the effect of stress on brain's distributed cortical information processing. It is apparent that, compared to more conventional correlation/coactivation based appraoches, the utilization of such measures of directed flow of information as TE can provide a more comprehensive veiw of the brain's regional interactivity and information sharing. Despite this intersting feature, TE suffers from one key criterion i.e., its expensive computation. Specifically, TE's time complexity is O(), where R refers to its computational resolution. In this respect, recent research presented substantial progress on addressing this limitation [120,121]. However, further analytical studies are necessary to allow for harnessing the full utility of TE for qunatification of the effect of various psychological and mental disorders on brain function. This is in particular crucial to enable the use of TE as a useful feature for real-time data-driven approaches to decoding of the brain activity [58-61].”
Reviewer’s Comment: -Having a nice schematic diagram in the draft would be really helpful. This definitely alleviates the difficulty of going to details of the techniques for the readers.
Author’s Response: A new section is added (Section 2.4. An Overview of the Participants' Selection and EEG Inclusion Process, lines 165-208, in the current version of manuscript). It summarizes the procedure through which the participants and their EEG channels for each of HIGH and LOW stress-susceptible groups, per EC and EO settings, are selected. This Section also includes a new schematic digram (Figure 2, page 6, in the current version of manuscript) that illustrates the steps involved in this process.
Reviewer’s Comment: - There have been a surge in the application of Machine Learning and Statistical framework to solve the similar problem focused in this paper. The authors are encouraged to include some of the recent papers in the introduction to give a good holistic overview about the existing techniques to general readers:
# "Estimating regional cerebral blood flow using resting-state functional MRI via machine learning." Journal of neuroscience methods 331 (2020): 108528.
# "Gene regulatory network state estimation from arbitrary correlated measurements." EURASIP Journal on Advances in Signal Processing 2018, no. 1 (2018): 1-10.
# "Machine learning: assessing neurovascular signals in the prefrontal cortex with non-invasive bimodal electro-optical neuroimaging in opiate addiction." Scientific reports 9, no. 1 (2019): 1-14.
Author’s Response: The following content along with the references suggested by the reviewer are added to Section Introduction, lines 100-104, in the current version of the manuscript.
“From a broader perspective, these results posit the use of TE as a potential diagnostic/prognostic tool in identification of the effect of stress on distributed brain networks that are involved in the brain responses to stress. This observation becomes more intriguing considering the recent surge in application of machine learning and statistical framework to decoding of the brain activity [58–61].”
In addition, the importance of applicability of TE in machine learning based approaches is further highlighted in Section Limitations and Future Direction, lines 560-570, in the current version of manuscript (please refer to the author’s response to the reviewer’s comment “Despite the fact that a good discussion about the superiority…”).
Reviewer’s Comment: - The format of some the references is not in standard form. These need to be checked and fixed.
Author’s Response: The current version of the manuscript uses the MDPI LateX template and follows the exemplars’ citations provided therein. However, the author would appreciate it if the reviewer could kindly specify any references whose formats are still incorrect so that they can be corrected. Thank you.

Reviewer 3 Report
The present manuscript describes a new multichannel EEG-based approach to characterize and classify stress susceptibility in humans. For this purpose the author made use of EEG- and psychological test data of 216 subjects as provided by the mind-body-brain dataset of the Leipzig Max Planck Institute. After selecting a subset of 20 (10+10) subjects showing the highest or lowest susceptability to stress (as reflected by their psychological test scores), the transfer entropy (TE) was used to analyse directed connectivity between all pairs of EEG signals. Subsequently these measures wer applied to physiologically characterize stress susceptibility. Contrasting both spatially cumulated and distributed TE values as observed in the two groups showed significant differences in a fronto-parietal network. In a leave one out validation procedure a 100% correct classification of the 20 subjects into the two groups was achieved.
Given these results the method may contribute to the understanding of stress susceptibility which still suffers from inconsistencies reported by other groups using a variety of modalities.
However, the current version of this manuscript gives rise to a number of questions and concerns:
Introduction; Page 2, line 49/50
ICA is a method to decompose multivariate data sets but not a measure of synchronized activity.
Introduction; Page 3, line 1/2
Wording: ‚Stress-susceptability’ is a feature whereas ‘change in flow of information’ reflects a difference between two or more conditions.
Methods; Page 2, line 113
Please specify the age distribution of the 162 participants subset
Methods; Page 4, line 147-158
From the text it cannot be derived how the number of participants exceeding/falling below the upper/lower bound of the item wise confidence interval (CI) was determined. By definition of the CI 2.5%/2.5% of all 122 participants (=3/3) should meet this criterion. However, the manuscript reports 72, 17, 56 etc. subjects.
Methods; Page 4, line 147-158
For the same reason it remains unclear how the 14 LOW and 26 HIGH stress susceptibility groups are defined along the test numbers.
Methods; Page 5, line 166-167
In the HIGH stress susceptible group 7 younger females and 1 older male were discarded. By which criterion? Can you rule out that this procedure introduces a bias regarding any comparison between the two groups?
Methods; Page 5, line 169-174
By taking the Euclidean distance between the test vectors, the numbers of the NEOFFI test (which are typically a magnitude below the other three test scores) will not have any influence on the resulting group following these distances. Consequently, there is a risk that the psychological structure of the resulting group is biased thereby potentially introducing a bias in the later results.
Methods; Page 5, line 193-195
Down sampling from 2500 Hz to 250 Hz and AFTERWARDS filtering with a 1-45 Hz bandpass (BP) may lead to aliasing errors in case of artifacts. Presumably, the BP filter was applied BEFORE down sampling – please check the info-file of the Leipzig data set.
Methods; Page 5, line 193-195
Was a zero phase BP-filter applied? Please check because otherwise the TE calculation could be corrupted in case of different spectral structures of the two time series.
Methods; Page 6, line 199-204
Principal component analysis (PCA) does not reduce the dimensionality (i.e. the number of EEG-channels) of the analysed data sets so that the sentence ‚the dimensionality ... was reduced by performing a PCA’ does not make sense.
Methods; Page 6, line 199-204
An ICA was applied to remove artifacts. But nothing is said about the criterion to define the number and selection of components to be removed.
Was the PCA and ICA conducted by the MPI Leipzig as part of their preprocessing (I could not finding any hint in their readme file) or was it done by own work?
Methods; Page 6, line 208-210
How can a PCA lead to ‚missing EEG channels’? PCA may well lead to decreased amplitude but not to a missing channel.
Methods; Page 7, line 223-228
Nothing is said about the algorithm to estimate the probabilities needed to calculate the TE.
Methods; Page 8, line 250-261
These two paragraphs are hard to read:
How were the CIs ‚obtained’ (line 250), were the CIs determined separately for each oft he 53x53 entries, how was the bootstrapping conducted (what does ‚simulation run’ mean?), why are ‚non-zero TE entries’ counted (exact zero TE entries are highly unlikely at all)?
Methods; Page 10, line 310-323
This paragraph is confusing and hard to read. Please re-phrase and – potentially - add a block diagram to illustrate the procedure.
Methods; Page 10, line 324-344
These two paragraphs are as well confusing and hard to follow.
Methods; Page 10, line 364-371
According to this paragraph all statistics done separately for the two groups are Bonferroni corrected for multiple (2) comparisons. However, the multiple tests reported in tables 7...10 seem not be corrected.
Results; Page 15, Figure 7
The maps nicely illustrate the structure of connectivity changes on a descriptive level. However, any statistical significance values are missing so that the reader cannot distinguish systematic from potentially random effects.
Author Response
First and foremost, the author would like to take this opportunity to express his gratitude for the reviewer’s time and kind consideration to review the present manuscript. The reviewer’s comments substantially improved the quality of the present study and its presentation.
In what follows, point-by-point responses to the reviewer’s comments and concerns are provided.
Sincerely,
Reviewer 3
Prior to responding to the reviewer’s comments, the author would like to clarify that the present study used the pre-processed EEG data that were provided by the Max Planck Institute Leipzig Mind-Brain-Body Dataset [53]. This choice was to increase the reproducibility of the results presented in the present manuscript. Upon receiving the comments by the reviewer and in order to provide further information on reported pre-processing steps in [53], author found another publication in PLOS Computational Biology that was coauthored by one of the authors in [53] and used the same EEG dataset (reference [70] in the present version of manuscript). Therefore, the author of the present manuscript used this latter publication to fill in some of the missing information with regards to pre-processing steps. Subsequently, reference [70] is cited accordingly and as appropriate.
Reviewer’s Comment: Introduction; Page 2, line 49/50
ICA is a method to decompose multivariate data sets but not a measure of synchronized activity.
Author’s Response: A number of previous studies used ICA as a measure of coactivation (e.g., references [16] and [27-29] in the current version of manuscript) among the brain’s regions. Although coactivation could imply synchronized activities among these regions, the author agrees with the reviewer’s comment and that the use of “synchrony” might potentially lead to misinterpretation. Therefore, all occurrences of “synchronized activity” in reference to ICA are replaced with “coactivation” in the current version of manuscript.
Reviewer’s Comment: Introduction; Page 3, line 1/2
Wording: ‚Stress-susceptability’ is a feature whereas ‘change in flow of information’ reflects a difference between two or more conditions.
Author’s Response: As a first step for selecting the participants for HIGH and LOW groups, the present study made use of individuals’ responses to questionnaires. The reference [53] from which the data in the present study is taken did not specifically categorize their participants as “stressed” and “non-stressed” individuals. Therefore, to avoid any undesirable generalization and overestimation, the author opted for “stress-susceptible” and “stress-susceptibility.” These phrases were meant to highlight the potentially stressed individuals in HIGH group while preventing any overemphasis. However, the author is willing to rephrase them if the reviewer considers such a change to be necessary. The author would also appreciate it if the reviewer could possibly suggest any alternatives. Thank you.
Reviewer’s Comment: Methods; Page 2, line 113
Please specify the age distribution of the 162 participants subset
Author’s Response: This information is provided in Section 2.1. Participants (lines 118-119, in the current version of manuscript) as follows.
“In the present study, we used this subset of 162 participants (i.e., 216 - 54 = 162 participants) (age: M = 38.61, Mdn = 30.0, SD = 20.14).”
Reviewer’s Comment: Methods; Page 4, line 147-158
From the text it cannot be derived how the number of participants exceeding/falling below the upper/lower bound of the item wise confidence interval (CI) was determined. By definition of the CI 2.5%/2.5% of all 122 participants (=3/3) should meet this criterion. However, the manuscript reports 72, 17, 56 etc. subjects.
Author’s Response: To address the reviewer’s comment, a new section is added (Section 2.4. An Overview of the Participants' Selection and EEG Inclusion Process, lines 165-208, in the current version of manuscript). This Section summarizes the procedure through which the participants and their EEG channels for each of HIGH and LOW stress-susceptible groups, per EC and EO settings, are selected. It also includes a new figure (Figure 2, page 6, in the current version of manuscript) that illustrates the steps involved in this process.
Reviewer’s Comment: Methods; Page 4, line 147-158
For the same reason it remains unclear how the 14 LOW and 26 HIGH stress susceptibility groups are defined along the test numbers.
Author’s Response: Please refer to the author’s response to the reviewer’s comment “Methods; Page 4, line 147-158From the text it cannot be derived how the number…”
Reviewer’s Comment: Methods; Page 5, line 166-167
In the HIGH stress susceptible group 7 younger females and 1 older male were discarded. By which criterion? Can you rule out that this procedure introduces a bias regarding any comparison between the two groups?
Author’s Response: As noted in the manuscript (Appendix A.1.1 Participants' Selection Based on Their Responses to NEO-FFI, PSQ, STAI-G-X2, lines 561-562), there were only one older male (age range, 60-65) in the HIGH and one older male (age range, 70-75) in the LOW stress-susceptible groups. Additionally, we also observed that there was only one younger female (age range, 30-35) in the LOW stress-susceptible group. These participants were discarded from further analyses for two reasons.
- First, the absence of younger female in HIGH group meant that the effect of gender on the observed results could not have been verified. This was an important issue, considering the findings that showed the differential effect of participants' gender on the brain response to stress (e.g., references [83] and [114] in the current version of manuscript).
- Second, we discarded the one older male in each of LOW and HIGH groups since their low sample could not have allowed for further analysis to verify whether age played a role in our results. This was also an important issue, given the effect of ageing on reduced complexity and information processing capacity of the brain (e.g., references [115-119] in the current version of manuscript).
To further clarify the reasons why these participants were excluded from the analyses, the following is added to Section Appendix A.1.1 Participants' Selection Based on Their Responses to NEO-FFI, PSQ, STAI-G-X2, lines 564-572, in the current version of manuscript.
“The absence of younger female participants in HIGH group (i.e., in addition to a single younger female in LOW group) meant that we were not able determine the effect of gender on potential differences betweeen these two groups. This was an important issue, considering the findings [82,114] that showed the differential effect of participants' gender on their brain response to stress. In the same vein, having only one older male in each of LOW and HIGH groups would have not allowed for further analysis to verify whether age played a role in our observations. This was also an important issue, given the effect of ageing on reduced complexity and information processing capacity of the brain [115-119]. Therefore, we decided to discard the older participants and the female participants from our further analyses.”
Furthermore, the following is included in Section Limitations and Future Direction, lines 469-480, in the current version of manuscript.
“While selecting the participants for the present study, the limited number of female participants in the final HIGH and LOW stress-susceptible groups left us with no choice but to exclude them from our further analyses. On the other hand, Lighthall et al. [114] and Seo et al. [82] showed the differential effect of participants' gender on the brain response to stress. This necessitates further investigation of our findings in settings in which both male and female (as well as more gender-diverse individuals) are included. Similarly, the limited number of older adults (i.e., one individual in each of LOW and HIGH stress-susceptible groups) required us to exclude them from further analyses since their limited sample would have not allowed us to verify whether the observed results were due to stress or the effect of ageing on reduced complexity and information processing capacity of the brain [115-119]. As a result, it is crucial to consider the sample population that comprise older people as well as adolescent [10,27] for drawing more informed conclusion on the change in cortical information processing in response to stress.”
Reviewer’s Comment: Methods; Page 5, line 169-174
By taking the Euclidean distance between the test vectors, the numbers of the NEOFFI test (which are typically a magnitude below the other three test scores) will not have any influence on the resulting group following these distances. Consequently, there is a risk that the psychological structure of the resulting group is biased thereby potentially introducing a bias in the later results.
Author’s Response: The author thanks the reviewer for raising this concern. This is indeed an important issue which could have affected the subsequent analyses. However, the author missed to note that the values of participants’ responses to all questionnaires were normalized first in the first draft of this manuscript. This missing information is corrected for in the current version of manuscript (lines 579-581) as follows.
“The element of these vectors were the participants' normalized responses to neuroticism (NEO-FFI), worries (PSQ), tension (PSQ), and STAI trait anxiety (STAI-G-X2).”
Reviewer’s Comment: Methods; Page 5, line 193-195
Down sampling from 2500 Hz to 250 Hz and AFTERWARDS filtering with a 1-45 Hz bandpass (BP) may lead to aliasing errors in case of artifacts. Presumably, the BP filter was applied BEFORE down sampling – please check the info-file of the Leipzig data set.
Author’s Response: The author checked both [53] as well as another publication that used the same dataset and was coauthored by one of the authors of [53] (reference [70] in the current version of the manuscript). Both publications reported applying BP after down sampling.
Reviewer’s Comment: Methods; Page 5, line 193-195
Was a zero phase BP-filter applied? Please check because otherwise the TE calculation could be corrupted in case of different spectral structures of the two time series.
Author’s Response: Although this information was not reported in [53], the author of the present manuscript found it in [70] which reported four-order in both directions to minimize zero-crossing by low-frequency drifts. This information is added to the current version of the manuscript as follows (Section 2.3. Preprocessing, lines 136-138).
“… It was then, per channel, per participant, bandpass-filtered within 1-45 Hz using an eight-order Butterworth filter (i.e., four-order in both directions to minimize zero-crossing distortions by low-frequency drifts [70]).”
Reviewer’s Comment: Methods; Page 6, line 199-204
Principal component analysis (PCA) does not reduce the dimensionality (i.e. the number of EEG-channels) of the analysed data sets so that the sentence ‚the dimensionality ... was reduced by performing a PCA’ does not make sense.
Author’s Response: [53] reported that “The dimensionality of the data was reduced using principal component analysis (PCA), by keeping PCs (N 30) that explain 95% of the total data variance.” Considering (N 30) reported by the authors in [53], it was not plausible to assume that N was referring to number of data points. Therefore, prior to the start of the present study, the author contacted the first author of [53] who then asked another coauthor to respond to his query. They confirmed that PCA was applied on the channels. The author of the present study also consulted the reference [70] to check whether any further information was available. Unfortunately, [70] did not report any further information.
Reviewer’s Comment: Methods; Page 6, line 199-204
An ICA was applied to remove artifacts. But nothing is said about the criterion to define the number and selection of components to be removed.
Author’s Response: With regard to ICA, [53] reported that Infomax (runica) algorithm was used. This information is added to Section 2.3. Preprocessing, lines 144-146, in the current version of manuscript. It reads as follows.
“This step was then followed by independent component analysis (ICA) on temporal (i.e., EEG channels' data points) dimension of data using the Infomax (runica) algorithm.”
[53] also reported that their analyses of EEG data were performed in EEGLAB (version 14.1.1b) for MATLAB (Delorme and Makeig, 2004). This information is added to Section 2.3. Preprocessing, lines 151-152, in the current version of manuscript. It reads as follows.
“These analyses were performed using EEGLAB [71] (version 14.1.1b) for MATLAB (Delorme and Makeig, 2004).”
Furthermore, [53] reported that the components that reflected eye movement, eye blink, or heartbeat related artifacts were removed and that the retained independent components for EO (M = 19.70, range = 9.0-30.0) and EC (M = 21.40, range = 14.0-28.0) conditions were back-projected to the sensor space for further analysis. In addition, [70] reported the following settings for ICA: step size: , annealing policy: when weight change > 0.000001, learning rate is multiplied by 0.98, stopping criterion maximum number of iterations 512 or weight change < 0.000001. This information is added to Section 2.3. Preprocessing, lines 144-148, in the current version of manuscript. It reads as follows.
“This step was then followed by independent component analysis (ICA) on temporal (i.e., EEG channels' data points) dimension of data using the Infomax (runica) algorithm (step size: , annealing policy: when weight change > 0.000001, learning rate was multiplied by 0.98, stopping criterion maximum number of iterations 512 or weight change < 0.000001).”
Reviewer’s Comment: Was the PCA and ICA conducted by the MPI Leipzig as part of their preprocessing (I could not finding any hint in their readme file) or was it done by own work?
Author’s Response: Yes, the present study used the preprocessed EEG data that were provided by Max Planck Institute Leipzig Mind-Brain-Body Dataset [53]. This is acknowledged in Section 2.2.2 Preprocessing, line 134, in the current version of manuscript, as follows.
“We used the preprocessed EEG recordings that were available through mind-body-brain dataset [53].”
Reviewer’s Comment: Methods; Page 6, line 208-210
How can a PCA lead to ‚missing EEG channels’? PCA may well lead to decreased amplitude but not to a missing channel.
Author’s Response: Please refer to the author’s response to the reviewer’s comment “Methods; Page 6, line 199-204Principal component analysis (PCA) does not reduce …”
Reviewer’s Comment: Methods; Page 7, line 223-228
Nothing is said about the algorithm to estimate the probabilities needed to calculate the TE.
Author’s Response: The author apologizes for missing this information. The present study used the TE implementation in [81]. This information is added to Section 2.5.4. TE Computation, Effect-Sizes and Bonferroni Correction, line 303, in the current version of manuscript. It reads as follows.
“We used the TE implementation in [81].”
Reviewer’s Comment: Methods; Page 8, line 250-261
These two paragraphs are hard to read:
How were the CIs ‚obtained’ (line 250), were the CIs determined separately for each oft he 53x53 entries, how was the bootstrapping conducted (what does ‚simulation run’ mean?), why are ‚non-zero TE entries’ counted (exact zero TE entries are highly unlikely at all)?
Author’s Response: Regarding the first part of the reviewer’s comment, please refer to the author’s response to the reviewer’s comment “Methods; Page 4, line 147-158 From the text it cannot be derived how the number…” With regard to zero TEs: TE = 0 indicates no transfer of information between two channels. In the context of present study, these entries were of two types
- entries whose computed values were originally zero. In other words, these were entries that corresponded to paired channels with no flow of information between them.
- entries whose values fell below the upper boundary of TEs’ 95.0% confidence interval (newly added Section 2.4. An Overview of the Participants' Selection and EEG Inclusion Process, lines 165-208, and Figure 2) and therefore discarded (i.e., their non-significant TEs were set to zero).
Reviewer’s Comment: Methods; Page 10, line 310-323
This paragraph is confusing and hard to read. Please re-phrase and – potentially - add a block diagram to illustrate the procedure.
Author’s Response: To more clearly represent the total TE computation, a new equation (equation (1), page 8, in the current version of manuscript) is added. Furthermore, the content of this paragraph is also modified to increase its readability (lines 240-250, in the current version of manuscript). The author apologizes for not including the modified paragraph here. This is due to the presence of the new equation and some additional symbols that may lose their formatting if incorporated in this review response.
Reviewer’s Comment: Methods; Page 10, line 324-344
These two paragraphs are as well confusing and hard to follow.
Author’s Response: This paragraph is modified as follows.
- First, the content related to Wilcoxon test was rewritten to increase its readability. Its current content reads as follows (lines 251-259)
“We then performed Wilcoxon rank sum test between each pair of selected channels for HIGH and LOW stress-susceptible groups (e.g., AF4 in HIGH and LOW). We carried out this analysis on EEG channels of both, EC (Figure A2 (A)) and EO (Figure A2 (B)) settings. These tests identified a number of EEG channels that showed significantly different total TEs between HIGH and LOW stress-susceptible participants. In the case of EC, these channels (9 EEG channels in total) were in frontal (FP2, AF4, F1, F6), centroparietal (CP3 and CP4), parietal (P2 and P3), and occipital (O2) regions. For EO (14 EEG channels in total), they were in frontal (AF4, F1, F4), frontotemporal (FT7), central (C1 and C6), centroparietal (CP3, CPZ, and CP4), parietal (P1 and PZ), parieto-occipital (PO7 and PO8), and occipital (O1) regions.”
- Next, the content that pertained to RSA and GLM were broken into two enumerated parts (lines 260-281) as follows.
“We used these EEG channels, per EC and EO settings, and performed two additional analyses on them, thereby evaluating their specificity and sensitivity to differentiate between HIGH and LOW stress-susceptible groups. These analyses were
- Representational Similarity Analysis (RSA) [73,74]: we applied RSA with Euclidean similarity distance on these channels. The input to RSA was the vectors of significantly different EEG channels of the participants (i.e., 1 X 9 and 1 X 14 vectors, per participant, for each of the EC and EO settings). We then performed Wilcoxon rank sum tests on HIGH and LOW RSA-based clusters (both within- and between-cluster). Next, we adapted a one-holdout cross-validation strategy to determine how well the individuals from each of these two groups could be predicted. For this purpose, we separated one of the participants from HIGH/LOW stress-susceptible group and computed the Euclidean similarity distances between all the remaining participants. We then predicted the group membership of the holdout participant by computing the vicinity of this participant’s vector to the center of HIGH and LOW RSA-based clusters. We repeated this step for every individual (i.e., 10 HIGH and 10 LOW participants), per EC and EO settings.
- General Linear Model (GLM) [75,76]: To realize the importance of each of these EEG channels, we opted for GLM analysis using logistic regression with sigmoid function. The inputs to this model were N X M matrices where N refers to the number of participants (i.e., 20 in our case) and M is the number of channels whose total TEs significantly differed between HIGH and LOW groups (i.e., 9 and 14 in the case of EC and EO, respectively). We used 1 and 0 as class labels for HIGH and LOW stress-susceptible groups, per EC and EO settings. We then carried out ANOVA analyses on this model’s coefficients (i.e., weights), per EC and EO settings. We used Matlab 2016a ("fitlm" and its corresponding "anova" functions) for these analyses.”
Reviewer’s Comment: Methods; Page 10, line 364-371
According to this paragraph all statistics done separately for the two groups are Bonferroni corrected for multiple (2) comparisons. However, the multiple tests reported in tables 7...10 seem not be corrected.
Author’s Response: The analyses reported in the current study included two groups: HIGH and LOW stress-susceptible groups. This meant that the Bonferroni-corrected p-value at 95.0% confidence interval was 0.05/2 = 0.025. In this regard, the author observed that p-values in Table 7, 9, and 10 were sufficiently small to make for a reliable margin from 0.025 corrected p-value. On the other hands, the p-values for GLM were substantially above corrected p-value. Therefore, the actual p-values were reported for better documentation of actual results. However, the author has no objection to modifying these p-values and reporting their corrected values in case the reviewer finds this change necessary. Thank you.
Reviewer’s Comment: Results; Page 15, Figure 7
The maps nicely illustrate the structure of connectivity changes on a descriptive level. However, any statistical significance values are missing so that the reader cannot distinguish systematic from potentially random effects.
Author’s Response: These maps in fact visualize the channels whose statistics were reported in Tables 4 and 5. Specifically, Figure 6 shows how the channels whose significant distributed TE differences were reported in these tables, had contributed to distributed cortical regions (i.e., as per EEG channels’ arrangements in [53]). Similarly, Figures 7 and 8 help identify where the target channels were. They also help realize the significantly reduced number of such target channels in the case of HIGH stress-susceptible group. With regard to this latter observation, the manuscript reports the Wilcoxon rank sum tests (i.e., HIGH vs. LOW) on the number of regions that each of these channels with significantly different distributed TEs transferred information to. These tests are included in lines 360-363 (EC setting) and 699-702 (EO setting) in the current version of manuscript.

Round 2
Reviewer 1 Report
The author has addressed most of my concerns.
My last comments can be summarized below:
1) The author should change the paragraph in the introduction
The goal of this study is threefold. We aim to show .....
and not like presenting your findings in the discussion part after
demonstrating them in the results section.
2) I insist that you cannot use features extracted from Wilcoxon Rank test as input to RSA.
If you want to separate them then you can use either the whole set of features in RSA or you can use a feature selection algorithm within
a cross-validation scheme like a 5-fold or one-holdout cross-validation.
If you select the features using the labels of groups or conditions outside a
discrimination analysis then you add a strong bias to your analysis.
You can apply feature selection approach within every round of
one-holdout cross-validation by applying it over the training sample.
You reported in section 2.5.2 - p.8:
we applied RSA with Euclidean similarity distance on these channels. The input to RSA was the vectors of significantly different EEG
channels of the participants (i.e., 1 x 9 and 1 x 14 vectors, per participant, for each of the EC and EO settings). We then performed Wilcoxon rank sum tests on HIGH and LOW RSA-based clusters (both within- and between-cluster). Next, we adapted a one-holdout cross-validation
strategy to determine how well the individuals from each of these two groups could be predicted
The sentence :
'We then performed Wilcoxon rank sum tests on HIGH and LOW RSA-based clusters (both within- and between-cluster). '
It is not need here. You have already selected both sets of features using Wilcoxon rank sum test.
Author Response
First and foremost, the author would like to take this opportunity to express his gratitude for the reviewer’s time and kind consideration to review the present manuscript. The reviewer’s comments substantially improved the quality of the present study and its presentation.
In what follows, point-by-point responses to the reviewer’s comments and concerns are provided.
Sincerely,
Reviewer 1
Reviewer’s Comment: 1) The author should change the paragraph in the introduction
The goal of this study is threefold. We aim to show .....
and not like presenting your findings in the discussion part after demonstrating them in the results section.
Author’s Response: This paragraph (Section Introduction, lines 86-93, in the current version of manuscript) provides a summary of the findings presented in this manuscript. This is to allow readers to have a clear and concise overview of the findings before reading through the details. To more clearly state this, it is rewritten in the current version of manuscript as follows (lines 86-93).
“The contribution of the present study is threefold. First, we identifies that the stress-susceptibility is charactrized by the change in flow of information in fronto-parietal brain network. Second, it verifies that these distributed fronto-parietal regions that expand bi-hemispherically are sufficient to significantly differentiate between the HIGH and LOW stress-susceptible groups. In this regards, it also indicates that although these regions contribute differently to such distinctions, their differential contributions are rather non-significant. Third, it shows that, in the case of HIGH stress-susceptible group, the flow of information between these distributed fronto-parietal brain regions is associated with a higher parietal-to-frontal flow of information.”
Reviewer’s Comment: 2) I insist that you cannot use features extracted from Wilcoxon Rank test as input to RSA.
If you want to separate them then you can use either the whole set of features in RSA or you can use a feature selection algorithm within
a cross-validation scheme like a 5-fold or one-holdout cross-validation.
If you select the features using the labels of groups or conditions outside a
discrimination analysis then you add a strong bias to your analysis.
You can apply feature selection approach within every round of
one-holdout cross-validation by applying it over the training sample.
Author’s Response: To avoid any potential misinterpretation of this part of the results by the readers, RSA analysis is removed from the present version of manuscript. Subsequently, the followings are removed from the current version of the manuscript.
- “with a significant specificity” is removed from Section 1. Introduction, line 89 (previous version of the manuscript).
- “We further verified that these regions were sufficient for robustly distinguishing between the HIGH and LOW stress-susceptible individuals.” Section 4 Discussion, 362-363 (previous version of the manuscript).
Reviewer’s Comment: You reported in section 2.5.2 - p.8:
we applied RSA with Euclidean similarity distance on these channels. The input to RSA was the vectors of significantly different EEG
channels of the participants (i.e., 1 x 9 and 1 x 14 vectors, per participant, for each of the EC and EO settings). We then performed Wilcoxon rank sum tests on HIGH and LOW RSA-based clusters (both within- and between-cluster). Next, we adapted a one-holdout cross-validation
strategy to determine how well the individuals from each of these two groups could be predicted
The sentence :
'We then performed Wilcoxon rank sum tests on HIGH and LOW RSA-based clusters (both within- and between-cluster). '
It is not need here. You have already selected both sets of features using Wilcoxon rank sum test.
Author’s Response: Please refer to the author’s response to the reviewer’s comment “2) I insist that you cannot use features extracted…”

Reviewer 2 Report
The paper is well revised and in my opinion is ready for publication.
Author Response
Thank you very much for your time and kind consideration to review this manuscript.
Reviewer 3 Report
The manuscript was substantially improved. However, the style of writing still is lacking clarity in several sections.
Author Response
First and foremost, the author would like to take this opportunity to express his gratitude for the reviewer’s time and kind consideration to review the present manuscript. The reviewer’s comments substantially improved the quality of the present study and its presentation.
In what follows, point-by-point responses to the reviewer’s comments and concerns are provided.
Sincerely,
Reviewer 3
Reviewer’s Comment: The manuscript was substantially improved. However, the style of writing still is lacking clarity in several sections.
Author’s Response: The reviewer’s comment is addressed as follows.
- Section 3. Distribution of the Information Transferred by the Channels with Significantly Different TEs: This part of the results is rewritten to more clearly present the correspondence of Figure 5 in the current version of manuscript (Caption: Eyes-Closed setting. Global distribution of TEs …) with the distributed TE results that preceded it. It reads as follows (lines 318-324).
“On the other hand, contribution of these channels' disributed TEs to other channels differed substantially between HIGH and LOW stress-susceptible groups. Figure 5 illustrates the distributed contribution of TEs from AF4, FP2, F1, F6, CP3, CP4, P2, P3, and O2 to the other channels whose significant differences were presented in Tables 4 and 5. This figure indicates that whereas transfer of information from these channels primarily contributed to the frontal regions in the case of HIGH stress-susceptible group (Figure 5 (A)), their contribution was more globally distributed among frontoparietal in the case of LOW stress-susceptible group (Figure 5 (B)).”
- This is followed by modifying the subsequent paragraph (lines 325-336, in the current version of manuscript) to more clearly associate Figure 6 and 7 (Captions: Eyes-Close (EC) setting. Transfer of information from … in the case of LOW stress-susceptible group. and Eyes-Close (EC) setting. Transfer of information from … in the case of LOW stress-susceptible group.) with the previous results related to distributed TEs. This paragraph reads as follows.
“Figures 6 and 7 visualize the cortical regions which each of these channels with significantly different distributed TEs transferred infromation to. More specifically, these figures show the directed transfer of information (i.e., TE) from (i.e., out-degree) AF4, FP2, F1, F6, CP3, CP4, P2, P3, and O2 to the other cortical regions in the case of HIGH and LOW stress-susceptible groups. A comparison between these figures clarifies the substantially frontal-oriented distributed TE among HIGH stress-susceptible group (figure 6) and its more distributed nature among frontoparietal regions in the case of LOW stress-susceptible individuals (figure 7). The more dense networks of TEs in the case of LOW versus HIGH stress-susceptible groups is apparent in these two figures. Wilcoxon rank sum test identified that the number of regions that each of these channels transferred information to was significantly higher in the case of LOW versus HIGH stress-susceptible participants (p = 0.000, W(16) = -3.59, r = 0.85, MHIGH = 13.11, SDHIGH = 1.36, MLOW = 33.78, SDLOW = 1.09). This difference was associated with a strong effect-size.”
- Similar changes were applied to Appendix B.3 Distribution of the Information Transferred by the Channels with Significantly Different Total TEs, lines 639-657, in the current version of manuscript.
- During the review process, the author realized that the results pertinent to RSA analysis was causing misinterpretation. Therefore, to avoid any potential misinterpretation by the readers, RSA analysis is removed from the present version of manuscript. Subsequently, the followings are removed from the current version of the manuscript.
- “with a significant specificity” is removed from Section 1. Introduction, line 89 (previous version of the manuscript).
- “We further verified that these regions were sufficient for robustly distinguishing between the HIGH and LOW stress-susceptible individuals.” Section 4 Discussion, 362-363 (previous version of the manuscript).
Additionally, Section 2.5.3 Distributed TEs (i.e., a subsection in 2.5. Analysis) was modified to reflect this modification. Its current content reads as follows.
“To realize the importance of each of these EEG channels in distinguishing between HIGH and LOW stress-susceptible groups, we performed General Linear Model (GLM) [75,76] analysis on them. We opted for GLM analysis using logistic regression with sigmoid function. The inputs to this model were N X M matrices where N refers to the number of participants (i.e., 20 in our case) and M is the number of channels whose total TEs significantly differed between HIGH and LOW groups (i.e., 9 and 14 in the case of EC and EO, respectively). We used 1 and 0 as class labels for HIGH and LOW stress-susceptible groups, per EC and EO settings. We then carried out ANOVA analyses on this model's coefficients (i.e., weights), per EC and EO settings. We used Matlab 2016a ("fitlm" and its corresponding "anova" functions) for these analyses. We reported the results of EC analyses in the main manuscript. We provided the results pertinent to EO setting in Appendix B.1 and Appendix B.2.”
- To more clearly summarize the contributions of the present study, its corresponding paragraph in Section 1 Introduction rewritten (lines 86-93, in the current version of manuscript) as follows.
“The contribution of the present study is threefold. First, we identifies that the stress-susceptibility is charactrized by the change in flow of information in fronto-parietal brain network. Second, it verifies that these distributed fronto-parietal regions that expand bi-hemispherically are sufficient to significantly differentiate between the HIGH and LOW stress-susceptible groups. In this regards, it also indicates that although these regions contribute differently to such distinctions, their differential contributions are rather non-significant. Third, it shows that, in the case of HIGH stress-susceptible group, the flow of information between these distributed fronto-parietal brain regions is associated with a higher parietal-to-frontal flow of information.”
Further to aforementioned changes and modifications, the entire manuscript was audited to ensure its readability as well as to fix any potential typos. However, the author would appreciate it if the reviewer can pinpoint any further changes/modification that are still necessary. Thank you.
